# Computational identification of epifriedelanol and derived analogs from *Mikania cordata* as potential HMG-CoA reductase inhibitors

**Miruna Banu, Sheikh Sunzid Ahmed, Momtaz Begum, M. Oliur Rahman**[ORCID]*

Department of Botany, Faculty of Biological Sciences, University of Dhaka, Dhaka, Bangladesh

* oliur.bot@du.ac.bd

## Abstract

Hypercholesterolemia, a major risk factor for cardiovascular diseases, arises from elevated blood cholesterol levels and remains a global health concern. The limitations of current therapies underscore the need for alternative drugs from natural sources. *Mikania cordata* (Asteraceae) is an ethnomedicinally important species that harbors numerous bioactive phytoconstituents. In this study, 91 phytocompounds of this medicinal species were virtually screened targeting the HMG-CoA (3-hydroxy-3-methylglutaryl-coenzyme A) reductase protein. Molecular docking, ADMET (absorption, distribution, metabolism, excretion, and toxicity), and MM/GBSA (molecular mechanics/generalized born surface area) analyses identified epifriedelanol as the best lead candidate among the phytocompounds with strong binding affinity (−8.6 kcal/mol), drug-likeness, and free binding energy (−39.5 kcal/mol), outperforming the standard drug atorvastatin (−7.7 kcal/mol and −21.4 kcal/mol). Analogs of epifriedelanol (EA) were further explored, generating 451 compounds. High-throughput screening of these analogs identified 244 compounds with a docking score higher than atorvastatin (−7.7 kcal/mol). The ADMET evaluation highlighted two analogs, EA2 and EA3, with docking scores of −9.3 kcal/mol and supportive MM/GBSA free energies (−31.9 and −43.7 kcal/mol). Molecular dynamics simulation (500 ns) confirmed the structural stability of epifriedelanol, EA2, and EA3, while essential dynamics and Gibbs free energy landscape analyses indicated a binding behavior comparable to that of atorvastatin. Target class analysis predicted interactions with nuclear receptors. These findings suggest that epifriedelanol and its analogs are promising natural leads against hypercholesterolemia, warranting further *in vitro* and *in vivo* validation.

## 1. Introduction

Over the last three decades, cardiovascular disease (CVD) has emerged as the leading contributor to the global burden of disease, accounting for 54% of mortality, 93% of prevalence, and 60% of disability-adjusted life years (DALYs) lost.

**Data availability statement:** All relevant data are within the manuscript and its supporting information files. Author generated code and raw data are available at Zenodo repository (https://doi.org/10.5281/zenodo.17553513).

**Funding:** The author(s) received no specific funding for this work.

**Competing interests:** The authors have declared that no competing interests exist.

Substantial disparities in outcomes persist both within and across regions, while CVD continues to impose an enormous economic burden, with annual healthcare expenditures estimated at USD 147 billion and productivity losses of approximately USD 216 billion [1,2]. A major factor driving this burden is hypercholesterolemia, a condition characterized by elevated blood cholesterol levels and recognized as a critical risk factor for CVD, including atherosclerosis, coronary artery disease, and stroke [3]. Increased low-density lipoprotein (LDL) cholesterol drives lipid plaque deposition, narrowing arteries, impairing blood flow, and triggering vascular inflammation, thereby increasing the risk of myocardial infarction and cerebrovascular events. The condition is further intensified by sedentary lifestyles, unhealthy diets, obesity, and type 2 diabetes, which collectively fuel the global rise in hypercholesterolemia [4,5]. According to the World Health Organization (WHO), high cholesterol is responsible for approximately 2.6 million deaths annually and contributes to nearly 29.7 million DALYs worldwide [6].

A key regulatory enzyme in cholesterol biosynthesis is 3-hydroxy-3-methylglutaryl-coenzyme A reductase (HMG-CoA reductase), which catalyzes the rate-limiting step of the mevalonate pathway leading to cholesterol production [7,8]. The excessive activity or expression of this enzyme is directly linked to elevated intracellular cholesterol levels. Consequently, HMG-CoA reductase represents a well-established and clinically validated therapeutic target, with statins–its competitive inhibitors, widely prescribed as lipid-lowering agents [9]. Although statins effectively reduce LDL-C levels and cardiovascular risk, their prolonged use is frequently accompanied by adverse effects, including hepatotoxicity and an increased risk of type 2 diabetes. These limitations highlight an unmet need for safer, naturally derived HMG-CoA reductase inhibitors that could serve as complementary or alternative therapies [10].

In this context, computational approaches have become indispensable for modern drug discovery, offering a rapid, cost-effective, and mechanistically insightful framework for identifying and optimizing new lead molecules [11,12]. Molecular docking enables the prediction and visualization of ligand-protein interactions at the atomic resolution, thereby facilitating the rapid screening of candidate molecules for their binding affinity to HMG-CoA reductase [13,14]. Beyond docking, molecular dynamics (MD) simulations offer time-resolved insights into the stability and conformational behavior of ligand–protein complexes under near-physiological conditions, helping to distinguish transient interactions from those with strong therapeutic potential [15]. Furthermore, MM/GBSA (molecular mechanics/generalized born surface area) free energy calculations refine candidate selection by estimating the thermodynamic stability and minimizing false positive results [16]. Together, these techniques form a rational pipeline for evaluating natural compounds prior to experimental validation.

Natural products have long served as a cornerstone of drug discovery, providing structurally diverse and biologically potent molecules that frequently act as lead compounds or direct therapeutic agents. Their inherent chemical diversity and evolutionary refinement make them invaluable resources for modern pharmaceuticals, particularly in oncology, infectious diseases, and cardiovascular therapy [17,18]. For instance, vinblastine and vincristine, two potent anticancer alkaloids, were originally

isolated from *Catharanthus roseus* and have been widely used for treating various malignancies, including Hodgkin's lymphoma and leukemia [19]. Similarly, artemisinin from *Artemisia annua* revolutionized malaria treatment, while paclitaxel from *Taxus brevifolia* became a mainstay in chemotherapy regimens [20,21]. These landmark discoveries underscore the potential of plant-derived metabolites to yield bioactive compounds with novel mechanisms of action and favorable safety profiles.

*Mikania cordata* (Burm.f.) B.L. Rob., a fast-growing tropical climber of the family Asteraceae, is native to South and Southeast Asia and has long been utilized in traditional medicine for its therapeutic benefits. Ethnomedicinally, different parts of the plant are used for diverse ailments. Extracts of the aerial parts act as spasmolytic agents; crushed leaves are applied to stop bleeding from cuts and wounds and to treat jaundice, septic sores, and snake bites; and infusions or decoctions of the leaves and flowers are traditionally used for colds, bronchitis, and diabetes [22–24]. Although these traditional applications primarily emphasize anti-inflammatory, antidiabetic, and wound-healing effects, they are functionally relevant to cardiovascular and metabolic health, as inflammation, oxidative stress, and diabetes are key contributors to atherosclerosis and dyslipidemia [25]. Notably, a closely related species, *Mikania micrantha*, has recently been reported to exhibit lipid- and blood pressure-lowering activities through enzymatic inhibition mechanisms, providing strong pharmacological evidence that members of the *Mikania* genus possess cardio-metabolic regulatory potential [26]. This evidence further supports the rationale for exploring *M. cordata* as a potential source of lipid-lowering bioactive compounds. In recent years, this species has garnered increasing pharmacological attention due to its broad spectrum of bioactivities. Phytochemical analyses have identified a range of bioactive constituents, including flavonoids, sesquiterpenes, steroids, and phenolic acids, many of which exhibit anti-inflammatory, antioxidant, antimicrobial, and lipid-lowering effects [27–29]. These observations warrant systematic investigation of *M. cordata* phytoconstituents for their potential to modulate lipid metabolism and cardiovascular pathways. However, despite these reports, no systematic study has explored whether the phytochemicals of *M. cordata* can interact with HMG-CoA reductase or modulate cholesterol biosynthesis. This knowledge gap provides a strong rationale for investigating its constituents through computational approaches.

To date, no studies have systematically explored the phytoconstituents of *M. cordata* for their potential to inhibit HMG-CoA reductase and modulate cholesterol metabolism. To address this knowledge gap, the present study employs a comprehensive computational drug discovery workflow, including high-throughput virtual screening, ADMET (absorption, distribution, metabolism, excretion, and toxicity) profiling, MD (molecular dynamics) simulations, and MM/GBSA (molecular mechanics/generalized born surface area) binding free energy analysis, to evaluate *M. cordata* phytocompounds as potential cholesterol-lowering agents. To the best of our knowledge, this study represents the first comprehensive computational assessment of *M. cordata* phytoconstituents for their potential to inhibit HMG-CoA reductase and modulate cholesterol metabolism.

## 2. Materials and methods

### 2.1. Protein preparation

The X-ray crystallographic structure of the human HMG-CoA reductase complexed with the inhibitor atorvastatin (PDB ID: 1HWK) was retrieved from the RCSB Protein Data Bank and selected as the structural template for receptor preparation [30]. The protein was processed and prepared using AutoDockTools v.1.5.6 [31] and SWISS-PDB Viewer v.4.10 [32], ensuring the removal of non-essential molecules and the optimization of structural integrity. Polar hydrogens were added and Kollman partial charges were assigned using AutoDockTools. The receptor structure was subsequently energy-minimized using the GROMOS96 43B1 force field in the SWISS-PDB Viewer to relieve steric clashes and obtain a low-energy conformation. Although explicit pKa calculations were not performed, this receptor preparation procedure follows established protocols commonly applied in recent protein-ligand docking studies [33–35]. Finally, the energy-minimized structure was converted from the PDB to the PDBQT format using OpenBabel v.3.1.1.1 to enable molecular docking analyses [36].

## 2.2. Preparation of ligands

Phytochemicals from *M. cordata* were identified and retrieved in 3D SDF format from the Indian Medicinal Plants, Phytochemistry and Therapeutics (IMPPAT) database [37]. Atorvastatin, used as the reference drug, was obtained from the PubChem database [38]. All ligand structures were energy-minimized using OpenBabel v3.1.1.1 with the MMFF94 force field and the steepest descent algorithm for 2000 steps to optimize their geometry prior to docking. The minimized structures were then converted to the PDBQT format to enable molecular docking analyses. While explicit protonation state adjustment at a defined pH was not performed, this preparation procedure provides standard hydrogen atom placement and optimized geometry for ligand docking, consistent with previously reported protocols [34,39,40].

## 2.3. Active site determination

For site-specific molecular docking, the active site of the receptor was identified using CASTp v.3.0 [41]. The PDB-formatted protein structure was analyzed to locate potential binding pockets, and the pocket with the largest surface area and volume was selected as the primary docking site.

## 2.4. Molecular docking

Based on the binding pocket predictions obtained from CASTp v.3.0, a docking grid box was configured with size dimensions of $52 \times 50 \times 50$ Å and center coordinates at 8.416 (X), 3.586 (Y), and 16.056 (Z) Å, ensuring complete coverage of the predicted active site cavity. A grid spacing of 0.375 Å was applied to define the resolution of the docking search space. This setup enabled the accurate simulation of ligand-receptor interactions.

To validate the docking protocol for the CASTp-predicted cavity, the co-crystallized ligand atorvastatin was used. The original predocking ligand conformation of atorvastatin was re-docked into the CASTp-defined binding pocket using the same docking parameters as those applied for the phytochemical screening. The re-docked pose was then aligned to the earlier pose (before docking) using PyMOL, and the RMSD between the two poses was calculated. This procedure tested the ability of the docking protocol to reliably reproduce ligand conformations within the CASTp-defined pocket.

Molecular docking was performed using EasyDock Vina v.2.237, a user-friendly docking tool for its flexibility, efficiency, and improved scoring accuracy [42]. The exhaustiveness parameter was set to 9 (fixed as the default for EasyDock Vina), ensuring sufficient sampling of the conformational space. The docking simulations identified favorable binding conformations and estimated the binding affinities of the selected phytocompounds and the target receptor. The scoring function of Vina, which combines empirical and knowledge-based potentials to estimate binding free energy in kcal/mol, was used to rank the docked poses [43]. Post-docking analysis was conducted using the Discovery Studio Visualizer, which provided detailed insights into non-covalent interactions, including hydrogen bonds, hydrophobic contacts, and π-π stacking [44].

## 2.5. ADMET properties evaluation

The pharmacokinetic and drug-likeness properties of the selected compounds were evaluated using the SwissADME server [45]. Subsequently, toxicity parameters were predicted separately using the STopTox server to assess potential adverse effects, completing the comprehensive ADMET evaluation [46].

## 2.6. Post-docking MM/GBSA free binding energy calculation

The MM/GBSA free binding energy of the top-ranked lead compounds was calculated using the Prime module of the Schrödinger Suite (v.2020-1). The computation employed the OPLS4 force field in combination with the VSGB solvation model to estimate the binding affinities of the ligand-receptor complexes [47].

                                                                                     

## 2.7. Analog-based drug design

The best lead compound was used to search for structural analogs through the Similar Structures Searching module of PubChem [48]. The retrieved analogs in the 3D SDF format were subjected to molecular docking using the same settings described in Section 2.4. Following docking, ADMET and MM/GBSA analyses were conducted using the parameters outlined in Sections 2.5 and 2.6, respectively, to identify the two most promising analogs as lead compounds.

## 2.8. Molecular dynamics (MD) simulation

To investigate the thermodynamic stability and dynamic behavior of the control drug, lead compounds, and apoprotein, molecular dynamics (MD) simulations were performed for 500 ns using the Desmond module of the Schrödinger v.2020-1 suite on Ubuntu v.22.04 [49]. Each ligand-receptor complex was solvated with the SPC water model within an orthorhombic periodic boundary box, ensuring a minimum distance of 10 Å between the protein surface and box edges. The OPLS4 force field was employed for system parameterization and energy minimization, with partial atomic charges automatically assigned during system preparation in Maestro to ensure full OPLS4 compatibility.

Prior to the production phase, the system underwent Desmond's default relaxation protocol, comprising Brownian dynamics under the NVT ensemble at 10 K with restrained solute heavy atoms (100 ps), subsequent short NVT and NPT equilibration stages (10 K, 12 ps each), pocket solvation, restrained NPT equilibration (12 ps), unrestrained NPT equilibration (24 ps), and finally, the unrestrained production simulation stage. This multi-step equilibration ensured gradual temperature and pressure stabilization before the 500 ns production run. Simulations were conducted under the NPT ensemble at a constant temperature of 300 K and pressure of 1.013 bar, maintained using the Nose–Hoover thermostat and isotropic pressure scaling, respectively. Long-range electrostatics were treated using the Particle Mesh Ewald (PME) method. Trajectories were saved at 500 ps intervals, generating approximately 1000 frames for post-simulation analysis, with energy measurements recorded every 1.2 ps to ensure precise thermodynamic evaluation.

The post-MD analyses included RMSD, RMSF, radius of gyration (Rg), and solvent-accessible surface area (SASA) to evaluate the conformational stability and compactness. Hydrogen bond dynamics and protein–ligand center-of-mass (COM) distance analyses were carried out using Schrödinger's "trajectory_analyze_hbonds.py" and "trajectory_asl_monitor.py" scripts, respectively, to assess non-covalent interaction persistence and ligand retention within the binding pocket. The use of the OPLS4 force field is justified by its superior torsional parameterization and improved accuracy in modeling non-bonded interactions relative to earlier OPLS versions. This has been demonstrated in recent studies employing the same target (1HWK) [50], where OPLS4 provided a reliable representation of the protein-ligand energetics and conformational behavior during long-timescale simulations.

## 2.9. Post-MD simulation MM/GBSA analysis

The MM/GBSA binding free energies were calculated for all 1000 frames of the 500 ns MD trajectories using Schrödinger's "thermal_mmgbsa.py" script [51]. The OPLS4 force field and VSGB solvation model were employed to estimate the free binding energy for each snapshot, providing a dynamic evaluation of the ligand-receptor interaction stability over the simulation. Average and per-frame binding energies were assessed to evaluate the temporal persistence of the binding affinities. To calculate the free binding energy, the following equation was used:

$$\Delta G(bind) \, = \, \Delta G(solv) \, + \, \Delta E(MM) \, + \, \Delta G(SA)$$

where $\Delta G(solv)$ represents the difference between the GBSA solvation energy of the protein-inhibitor complex and the sum of the solvation energies of the unliganded protein and inhibitor; $\Delta E(MM)$ denotes the difference in minimized energies between the complex and the unbound components; and $\Delta G(SA)$ represents the difference in surface area energies of the complex and the sum of the corresponding energies for the unliganded protein and inhibitor.

 

## 2.10. Principal component analysis and the Gibbs free energy landscape (FEL) study

Principal component analysis (PCA) was performed using the RMSD and Rg coordinates obtained from the MD simulation trajectories. The PCA module of the Statistics Kingdom server was used to complete the analysis [51]. The covariance matrix was selected keeping two dimensions and standardized scaling. For the Gibbs free energy landscape (FEL) analysis, a Python script was executed on Ubuntu Focal Fossa v.20.04.6 LTS. A 2D histogram of the PCA results was generated to estimate the probability distribution of conformational states, and Gibbs free energy values were calculated using Boltzmann statistics [51].

## 2.11. Evaluation of the drug target class

Drug target class prediction was performed using the SwissTargetPrediction server to evaluate the potential pharmacological targets of the selected lead candidates [52]. Canonical SMILES representations of the lead compounds were retrieved and submitted to the server. The *Homo sapiens* database was selected as the reference organism to ensure the relevance of the predicted targets to human biology.

## 3. Results

### 3.1. Molecular docking analysis

The CASTp server was used to predict the potential binding sites of the target protein for subsequent molecular docking analysis that revealed multiple surface pockets and cavities based on the geometric and topological parameters. Among these, the cavity ranked first (Rank 1) was selected as the most probable binding site due to its superior surface area and volume (S1 Fig). Specifically, the Rank 1 cavity exhibited a solvent-accessible surface area of 448.619 Å² and a total volume of 5080.485 Å³, making it the most prominent and structurally favorable site for ligand accommodation. These parameters suggest sufficient spatial dimensions to facilitate stable interactions with potential ligands. Thus, the Rank 1 cavity was used as the docking site for all the molecular docking simulations. The reliability of the docking setup was validated by re-docking atorvastatin into the CASTp-predicted cavity using the same parameters as for the phytochemical docking. The redocked pose was aligned with the original PubChem conformation in PyMOL, yielding an RMSD of 2.163 Å (Fig 1). This close agreement confirmed the accuracy and reproducibility of the docking protocol within the CASTp-defined pocket.

Virtual screening of 91 phytochemical constituents (S2 Fig) of *M. cordata* against HMG-CoA reductase revealed a broad spectrum of binding affinities, reflecting the structural diversity of the compounds. Among them, taraxasterol (IMPHY015081) exhibited the strongest binding affinity with a docking score of −8.8 kcal/mol, followed by epifriedelanol (−8.6 kcal/mol) and friedelin (−8.1 kcal/mol) (S1 Table). These three triterpenoids showed higher docking scores than the reference drug atorvastatin (−7.7 kcal/mol), indicating a potentially stronger and more stable interaction with the active site of HMG-CoA reductase (Table 1, Fig 2). In contrast, the compound 2-Hexenal (IMPHY011562) displayed the weakest interaction, with a docking score of −3.7 kcal/mol, reinforcing the wide variation in the binding affinities across the compound library.

### 3.2. Molecular interaction analysis

Molecular interaction analysis identified key binding site residues and the types of interactions formed between the top-ranked phytocompounds and HMG-CoA reductase, compared with the reference drug atorvastatin (Table 2). Taraxasterol formed a single hydrogen bond with Ala768 (2.42 Å) and engaged in hydrophobic contacts with Tyr517, Ile536, Ala768, and Pro813, contributing to its strong binding affinity (−8.8 kcal/mol). Similarly, epifriedelanol formed one hydrogen bond with Cys817 (3.23 Å) and interacted hydrophobically with Cys527, Val538, Ala556, Ala763, and Pro813, resulting in a docking score of −8.6 kcal/mol. In contrast, friedelin did not form hydrogen bonds and relied solely on hydrophobic interactions with Ile762, which may account for its slightly lower binding affinity (−8.1 kcal/mol). The control drug

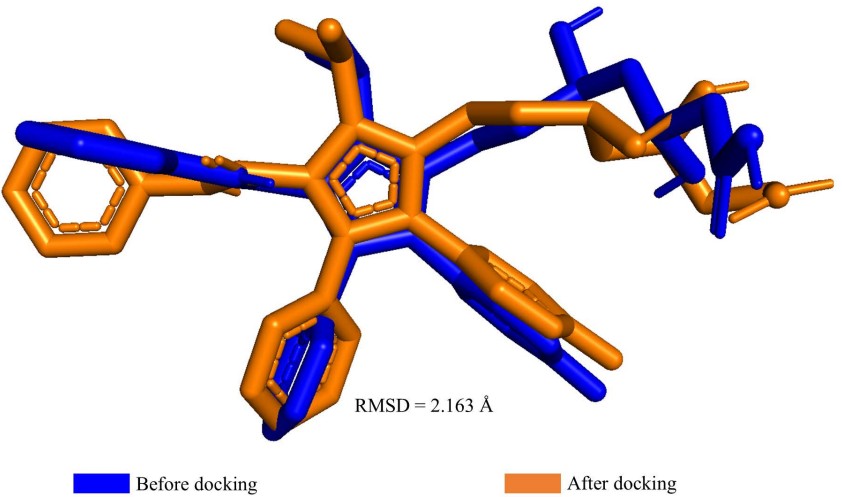

RMSD = 2.163 Å

Before docking After docking

**Fig 1. Validation of the molecular docking protocol.**

**Table 1. Virtual screening results of the top 24 phytocompounds of *M. cordata* targeting the HMG-CoA reductase protein.**

| Sl. No. | IMPPAT ID/PubChem CID | Name | Chemical formula | Parts used | Molecular weight (g/mol) | Affinity (kcal/mol) |
|---|---|---|---|---|---|---|
| 1 | IMPHY015081 | Taraxasterol | $C_{30}H_{50}O$ | Leaf | 426.73 | −8.8 |
| 2 | IMPHY011859 | Epifriedelanol | $C_{30}H_{52}O$ | Root | 428.75 | −8.6 |
| 3 | IMPHY011688 | Friedelin | $C_{30}H_{50}O$ | Root | 426.73 | −8.1 |
| 4 | IMPHY014842 | Stigmasterol | $C_{29}H_{48}O$ | Leaf | 412.7 | −7.4 |
| 5 | IMPHY001852 | Mikanin | $C_{18}H_{16}O_7$ | Stem | 344.32 | −7.3 |
| 6 | IMPHY001018 | Dihydromikanolide | $C_{15}H_{16}O_6$ | Leaf | 292.29 | −7.0 |
| 7 | IMPHY002144 | Deoxymikanolide | $C_{15}H_{16}O_5$ | Leaf | 276.29 | −7.0 |
| 8 | IMPHY006921 | Miscandenin | $C_{15}H_{14}O_5$ | Leaf | 274.27 | −6.9 |
| 9 | IMPHY014836 | Beta-sitosterol | $C_{29}H_{50}O$ | Leaf | 414.72 | −6.9 |
| 10 | IMPHY000982 | Mikanolide | $C_{15}H_{14}O_6$ | Stem | 290.27 | −6.7 |
| 11 | IMPHY002835 | Dihydroscandenolide | $C_{17}H_{20}O_7$ | Leaf | 336.34 | −6.7 |
| 12 | IMPHY010420 | Scandenolide | $C_{17}H_{18}O_7$ | Leaf | 334.32 | −6.7 |
| 13 | IMPHY011837 | Quinine | $C_{20}H_{24}N_2O_2$ | Leaf | 324.42 | −6.6 |
| 14 | IMPHY013080 | Alpha-calacorene | $C_{15}H_{20}$ | Leaf | 200.32 | −6.5 |
| 15 | IMPHY007646 | Anhydroscandenolide | $C_{15}H_{14}O_5$ | Leaf | 274.27 | −6.4 |
| 16 | IMPHY007840 | Spathulenol | $C_{15}H_{24}O$ | Leaf | 220.36 | −6.3 |
| 17 | IMPHY011564 | Germacra-1(10),5-dien-4-ol | $C_{15}H_{26}O$ | Leaf | 222.37 | −6.3 |
| 18 | IMPHY011542 | Beta-eudesmol | $C_{15}H_{26}O$ | Leaf | 222.37 | −6.2 |
| 19 | IMPHY011938 | Gamma-eudesmol | $C_{15}H_{26}O$ | Leaf | 222.37 | −6.2 |
| 20 | IMPHY011659 | Alpha-muurolene | $C_{15}H_{24}$ | Leaf | 204.36 | −6.1 |
| 21 | IMPHY011749 | Humulene epoxide II | $C_{15}H_{24}O$ | Leaf | 220.36 | −6.1 |
| 22 | IMPHY011793 | (+)−Gamma-cadinene | $C_{15}H_{24}$ | Leaf | 204.36 | −6.1 |
| 23 | IMPHY012667 | Caryophyllene oxide | $C_{15}H_{24}O$ | Leaf | 220.36 | −6.1 |
| 24 | IMPHY014906 | Cedrelanol | $C_{15}H_{26}O$ | Leaf | 222.37 | −6.1 |
| **60823 (Control drug)** | | Atorvastatin | $C_{33}H_{35}FN_2O_5$ | N/A | 558.65 | **−7.7** |

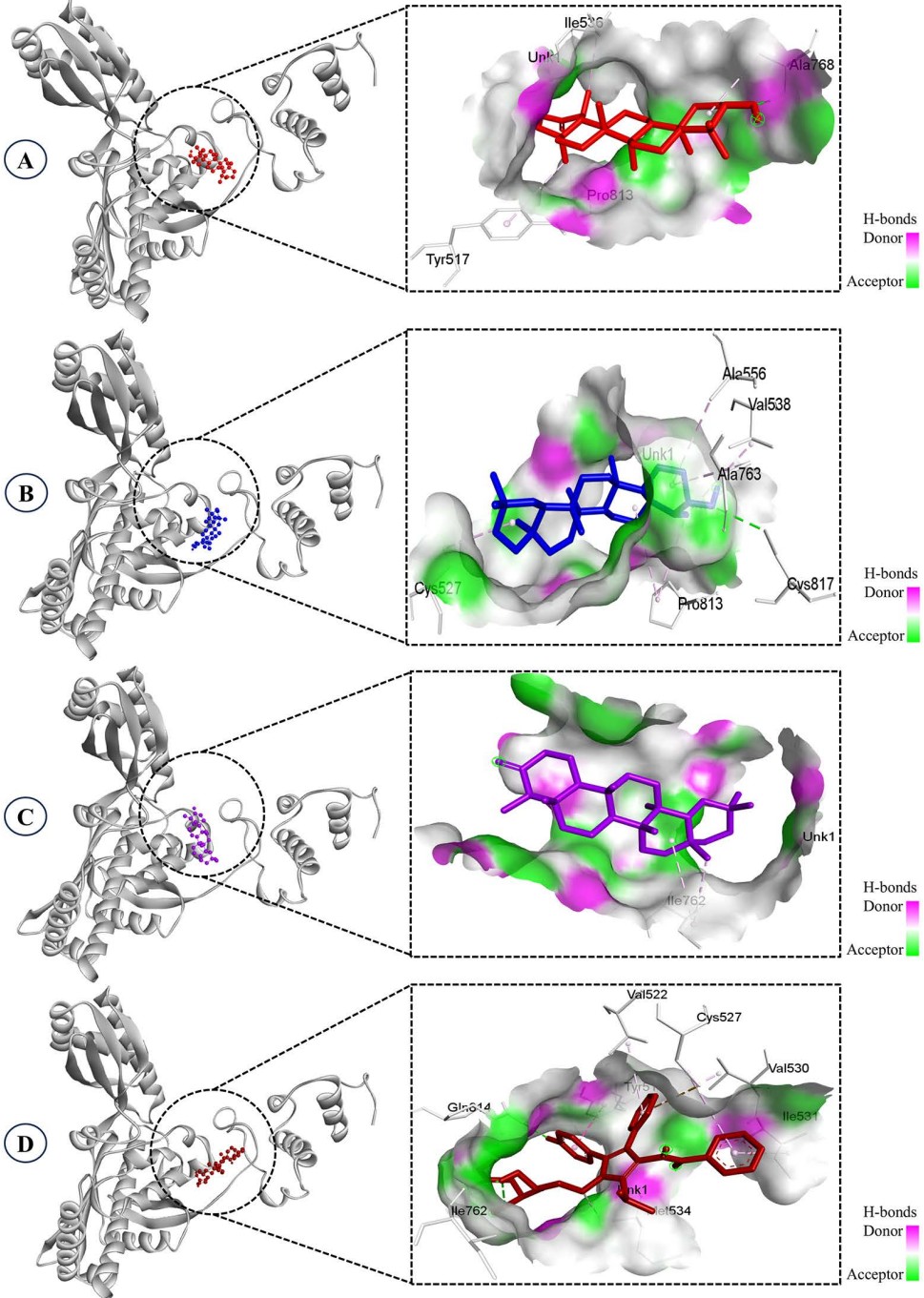

**Fig 2. Docked complexes showing 3D interactions of the ligands. A.** Taraxasterol, **B.** Epifriedelanol, **C.** Friedelin, **D.** Atorvastatin (control).

atorvastatin demonstrated a more complex interaction pattern, forming three hydrogen bonds with Ile762 and Gln814, along with hydrophobic contacts with Tyr517, Val522, Cys527, Val530, Ile531, and Met534, yielding a binding affinity of −7.7 kcal/mol. Notably, several key residues, such as Tyr517, Cys527, and Ile762, were shared between the selected

**Table 2. Molecular interaction assessment between the selected compounds of *M. cordata* and atorvastatin drug after molecular docking analysis.**

| Ligands | Binding sites | Hydrogen-bonding residues (Distance in Å) | Hydrogen bonds number | Hydrophobic-interaction | Binding affinity (kcal/mol) |
|---|---|---|---|---|---|
| Taraxasterol | Tyr517, Ile536, Ala768, Pro813 | Ala768[(2.42)] | 1 | Tyr517, Ile536, Ala768, Pro813 | −8.8 |
| Epifriedelanol | Cys527, Val538, Ala556, Ala763, Cys817, Pro813 | Cys817[(3.23)] | 1 | Cys527, Val538, Ala556, Ala763, Pro813 | −8.6 |
| Friedelin | Ile762 | None | 0 | Ile762 | −8.1 |
| Atorvastatin (control) | Tyr517, Val522, Cys527, Val530, Ile531, Met534, Ile762, Gln814 | Ile762[(2.06)], Gln814[(1.88, 2.42)] | 3 | Tyr517, Val522, Cys527, Val530, Ile531, Met534 | −7.7 |

phytocompounds and atorvastatin, indicating overlapping binding regions within the active site (Fig 3). This overlap suggests that these phytocompounds may bind in a manner similar to the reference inhibitor, potentially interfering with the enzyme's function through a comparable mechanism.

### 3.3. ADMET evaluation

The ADMET properties of the top three compounds—taraxasterol, epifriedelanol, and friedelin, were evaluated and compared with the reference drug atorvastatin (Table 3). All three phytocompounds exhibited comparable drug-likeness profiles to the control drug atorvastatin, supporting their potential as promising therapeutic candidates (S3 Fig).

All three compounds exhibited favorable physicochemical properties, including acceptable molecular weights (426.72–428.73 g/mol), low numbers of hydrogen bond donors (0–1) and acceptors (1), and very low topological polar surface areas (TPSA < 21 Å²), suggesting good passive membrane permeability. Lipophilicity was high, with consensus LogP scores ranging from 7.11 to 7.44, indicating strong hydrophobicity, in contrast to atorvastatin's more moderate value (4.94). Gastrointestinal (GI) absorption was predicted to be low for all compounds, including atorvastatin, likely due to their high lipophilicity and low TPSA values. None of the phytochemicals were predicted to inhibit key cytochrome P450 isoforms (CYP1A2, CYP2C9, or CYP2C19), whereas atorvastatin showed potential CYP2C19 inhibition, implying a lower risk of drug–drug interactions for the phytochemicals. All three leads were predicted to be poorly soluble in water. With respect to drug-likeness, all compounds exhibited one Lipinski rule violation and similar bioavailability scores (~0.55), indicating moderate oral bioavailability. No PAINS (Pan Assay Interference Compounds) alerts were detected, and the synthetic accessibility scores ranged from 5.17 to 5.40, suggesting moderate synthetic complexity. Toxicity predictions showed no risks for acute oral, inhalation, or dermal toxicity for any compound, although friedelin was flagged for potential skin sensitization and atorvastatin for possible eye irritation.

### 3.4. MM/GBSA free binding energy calculation

Based on the molecular interaction results and favorable ADMET properties, the top two phytochemicals from *M. cordata*—epifriedelanol and taraxasterol—were subjected to MM/GBSA free binding energy calculations to evaluate their binding stability with the target protein (S2 Table).

The study revealed that both compounds exhibited higher binding affinities compared with the reference drug atorvastatin. Epifriedelanol showed the most favorable binding free energy ($\Delta G$ Bind = −52.3 kcal/mol), followed by taraxasterol (−48.6 kcal/mol), while atorvastatin demonstrated a comparatively lower binding affinity (−44.7 kcal/mol). Notably, the van der Waals ($\Delta G$ vdW) interactions contributed significantly to the overall binding energy in all three complexes, particularly for epifriedelanol (−44.5 kcal/mol) and taraxasterol (−43.6 kcal/mol), indicating strong hydrophobic interactions within the binding pocket. In addition, the lipophilic component ($\Delta G$ Lipo) was substantial in epifriedelanol (−20.9

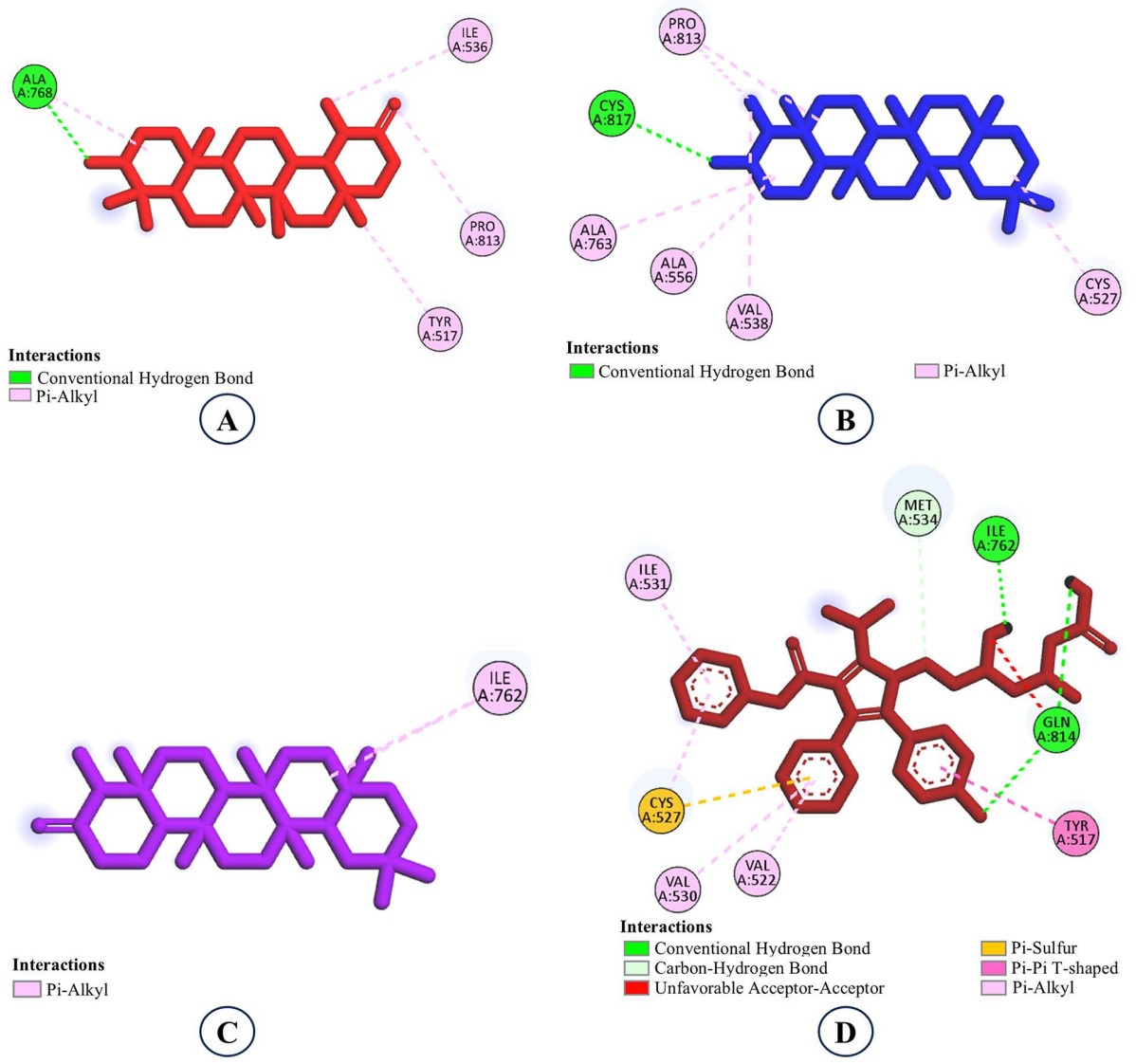

**Fig 3. Two-dimensional molecular interaction analysis of the top selected compounds and control drug. A.** Taraxasterol, **B.** Epifriedelanol, **C.** Friedelin, **D.** Atorvastatin (control).

kcal/mol) and taraxasterol (−17.9 kcal/mol), supporting the notion that hydrophobic residues in the binding site facilitate tight ligand accommodation. In contrast, atorvastatin's strong lipophilic interaction (−27.9 kcal/mol) was offset by its high solvation penalty (ΔG Bind Solv GB = 22.9 kcal/mol), reducing its overall binding strength. Moreover, atorvastatin's coulombic and covalent contributions (3.6 and 5.1 kcal/mol, respectively) suggest a greater reliance on electrostatic and covalent interactions, although these were insufficient to counteract the solvation loss. Overall, the binding energy profiles indicate that epifriedelanol and taraxasterol form more stable and energetically favorable complexes with the target enzyme compared with atorvastatin, underscoring their potential as promising natural HMG-CoA reductase inhibitors from *M. cordata*.

**Table 3. ADMET profile assessment of the top selected phytochemicals and control drug atorvastatin.**

| Parameters | Molecule | Taraxasterol | Epifriedelanol | Friedelin | Atorvastatin |
|---|---|---|---|---|---|
| Physicochemical properties | Formula | $C_{30}H_{50}O$ | $C_{30}H_{52}O$ | $C_{30}H_{50}O$ | $C_{33}H_{35}FN_2O_5$ |
| | Molecular weight (g/mol) | 426.72 | 428.73 | 426.72 | 558.64 |
| | H-bond acceptors | 1 | 1 | 1 | 6 |
| | H-bond donors | 1 | 1 | 0 | 4 |
| | Molar refractivity | 135.14 | 135.36 | 134.39 | 158.26 |
| | TPSA | 20.23 Å$^2$ | 20.23 Å$^2$ | 17.07 Å$^2$ | 111.79 |
| Lipophilicity | iLOGP | 4.68 | 4.84 | 4.49 | 3.58 |
| | XLOGP3 | 9.13 | 10.08 | 9.80 | 4.96 |
| | WLOGP | 8.02 | 8.25 | 8.46 | 6.54 |
| | MLOGP | 6.92 | 7.07 | 6.92 | 3.48 |
| | Silicos-IT Log P | 6.81 | 6.93 | 7.52 | 6.15 |
| | Consensus Log P | 7.11 | 7.43 | 7.44 | 4.94 |
| Pharmacokinetics | GI absorption | Low | Low | Low | Low |
| | CYP1A2 | No | No | No | No |
| | CYP2C19 | No | No | No | Yes |
| | CYP2C9 | No | No | No | No |
| | Log Kp | −2.42 cm/s | −1.76 cm/s | −1.94 cm/s | −6.19 cm/s |
| Water solubility (ESOL) | Log S | −8.24 | −8.85 | −8.66 | −5.99 |
| | Solubility (mg/ml) | 2.47E−06 | 6.08E−07 | 9.34E−07 | 5.78E−04 |
| | Solubility (mol/l) | 5.79E−09 | 1.42E−09 | 3.56E−08 | 1.03E−06 |
| | Class | Poorly soluble | Poorly soluble | Poorly soluble | Moderately soluble |
| Drug likeness | Lipinski (violations) | 1 | 1 | 1 | 1 |
| | Bioavailability score | 0.55 | 0.55 | 0.55 | 0.56 |
| Medicinal chemistry | PAINS (alerts) | 0 | 0 | 0 | 0 |
| | Synthetic accessibility | 5.40 | 5.27 | 5.17 | 4.95 |
| Toxicity | Acute inhalation | No | No | No | No |
| | Skin irritation and corrosion | No | No | No | No |
| | Eye irritation and corrosion | No | No | No | Yes |
| | Acute dermal | No | No | No | No |
| | Skin sensitization | No | No | Yes | No |
| | Acute oral | No | No | No | No |

### 3.5. Analog-based drug design

**3.5.1. High-throughput virtual screening of the structural analogs of epifriedelanol.** Epifriedelanol was identified as the best lead compound based on its highest MM/GBSA binding free energy score, demonstrating strong predicted binding stability with the target. To further optimize this lead, a high-throughput virtual screening (HTVS) was performed on a library of 451 structural analogs designed to explore the chemical space around epifriedelanol. From the 451 analogs, the top 20 compounds with docking scores of <−9.0 kcal/mol are presented in Table 4. Docking results for these analogs revealed binding affinities ranging from −5.0 to −9.4 kcal/mol, indicating varying degrees of interaction strength with the target protein. Notably, the top analogs—EA1 (−9.4 kcal/mol), EA2 (−9.3 kcal/mol), and EA3 (−9.3 kcal/mol) exhibited improved docking scores compared with the parent compound (−8.6 kcal/mol). These enhanced affinities suggest that the top analogs have the potential for stronger or more favorable interactions within the binding pocket, positioning them as promising candidates for further computational and experimental validation. Thus, the HTVS approach effectively

**Table 4. Molecular docking analysis of the top 20 structural analogs of epifriedelanol.**

| SL. No. | Analogs ID (EA*) | PubChem CID | Chemical Formula | Molecular Weight (g/mol) | Docking score (kcal/mol) |
|---|---|---|---|---|---|
| 1 | EA1 | 162857362 | $C_{30}H_{52}O_2$ | 444.7 | −9.4 |
| 2 | **EA2** | **71473698** | $C_{30}H_{52}O_3$ | 460.7 | −9.3 |
| 3 | **EA3** | **101528287** | $C_{30}H_{52}O_3$ | 460.7 | −9.3 |
| 4 | EA4 | 162934032 | $C_{30}H_{52}O_2$ | 444.7 | −9.3 |
| 5 | EA5 | 163000526 | $C_{30}H_{52}O_2$ | 444.7 | −9.3 |
| 6 | EA6 | 22294659 | $C_{30}H_{52}O_2$ | 444.7 | −9.2 |
| 7 | EA7 | 40557056 | $C_{30}H_{52}O$ | 428.7 | −9.2 |
| 8 | EA8 | 87055239 | $C_{30}H_{52}O$ | 428.7 | −9.2 |
| 9 | EA9 | 101124566 | $C_{31}H_{54}O$ | 442.8 | −9.2 |
| 10 | EA10 | 101124567 | $C_{31}H_{54}O$ | 442.8 | −9.2 |
| 11 | EA11 | 163032720 | $C_{31}H_{54}O$ | 442.8 | −9.2 |
| 12 | EA12 | 10478653 | $C_{30}H_{52}O_2$ | 444.7 | −9.1 |
| 13 | EA13 | 14466333 | $C_{30}H_{52}O$ | 428.7 | −9.1 |
| 14 | EA14 | 22294660 | $C_{30}H_{52}O_2$ | 444.7 | −9.1 |
| 15 | EA15 | 98520107 | $C_{30}H_{50}O$ | 426.7 | −9.1 |
| 16 | EA16 | 101603277 | $C_{30}H_{52}O_2$ | 444.7 | −9.1 |
| 17 | EA17 | 162934033 | $C_{30}H_{52}O_2$ | 444.7 | −9.1 |
| 18 | EA18 | 163043833 | $C_{30}H_{52}O_3$ | 460.7 | −9.1 |
| 19 | EA19 | 163190073 | $C_{30}H_{52}O_2$ | 444.7 | −9.1 |
| 20 | EA20 | 165358510 | $C_{30}H_{52}O_2$ | 444.7 | −9.1 |
|  | Atorvastatin |  | $C_{33}H_{35}FN_2O_5$ | 558.65 | −7.7 |

*EA: Control Epifriedelanol Analogs

prioritized analogs with superior binding profiles, accelerating the lead optimization process by identifying structurally related compounds with potentially improved efficacy.

**3.5.2. Molecular interaction analysis of structural analogs of epifriedelanol.** EA2 and EA3 were identified as the best analogs based on comprehensive ADMET profiling, which confirmed their favorable pharmacokinetic and toxicity properties, making them suitable for further investigation. These analogs were subsequently subjected to detailed molecular interaction studies to elucidate their binding modes with the target protein (Table 5).

Interaction analysis revealed that both EA2 and EA3 occupied the same hydrophobic cavity of the target protein and interacted with a common group of residues including Tyr533, Ile536, Val538, Ala556, Ala763, and Pro813, mainly through pi–alkyl and carbon–hydrogen interactions (Figs 4 and 5). For EA2, strong pi–alkyl contacts were observed with Tyr533 (4.94 Å), Ile536 (5.08 Å), Val538 (5.14 Å), Ala556 (4.68 Å), and Pro813 (4.41 Å), while a short carbon–hydrogen inter-action was formed with Ala763 (3.18 Å). The analog EA3 displayed a nearly identical interaction profile, forming pi–alkyl contacts with Tyr533 (4.88 Å), Ile536 (5.06 Å), Val538 (5.12 Å), Ala556 (4.70 Å), and Pro813 (4.45 Å), along withS5 a carbon–hydrogen interaction with Ala763 (3.16 Å). Remarkably, neither analog formed hydrogen bonds with the protein, as reflected by the absence of hydrogen-bonding residues. Despite this, both compounds exhibited strong binding affini-ties (−9.3 kcal/mol), indicating highly favorable and stable binding within the binding pocket. In contrast, the control drug atorvastatin interacted with a different subset of residues, including Tyr517, Val522, Cys527, Val530, Ile531, Met534, Ile762, and Gln814. Importantly, atorvastatin formed three hydrogen bonds involving Ile762 and Gln814, with bond lengths ranging from 1.88 to 2.42 Å, reflecting specific polar interactions that typically contribute to binding specificity and stability.

**Table 5. Molecular interaction assessments of the EA analogs and atorvastatin.**

| Ligands | Binding sites | Hydrogen-bonding residues (Distance in Å) | Hydrogen bonds number | Hydrophobic-interaction | Docking score (kcal/mol) |
|---|---|---|---|---|---|
| EA2 | Tyr533, Ile536, Val538, Ala556, Ala763, Pro813 | None | 0 | Tyr533, Ile536, Val538, Ala556, Ala763, Pro813 | −9.3 |
| EA3 | Tyr533, Ile536, Val538, Ala556, Ala763, Pro813 | None | 0 | Tyr533, Ile536, Val538, Ala556, Ala763, Pro813 | −9.3 |
| Atorvastatin (control) | Tyr517, Val522, Cys527, Val530, Ile531, Met534, Ile762, Gln814 | Ile762[(2.06)], Gln814[(1.88, 2.42)] | 3 | Tyr517, Val522, Cys527, Val530, Ile531, Met534 | −7.7 |

*EA: Epifriedelanol Analogs

These findings underscore a key distinction in the binding mechanisms: while atorvastatin relies on a balanced combination of hydrophobic contacts and hydrogen bonding, EA2 and EA3 achieve superior binding affinity predominantly through extensive hydrophobic interactions with multiple residues lining the active site. This suggests that the hydrophobic contacts formed by EA2 and EA3 play a pivotal role in stabilizing the ligand-protein complex, potentially leading to stronger inhibitory effects. The absence of hydrogen bonds does not appear to diminish their binding strength, underscoring the significance of the nonpolar contacts in these complexes.

**3.5.3. ADMET analysis of the structural analogs.** The two lead analogs, EA2 and EA3, are stereoisomers, and their ADMET properties were compared with the control drug atorvastatin to evaluate their drug-likeness and toxicity profiles (Table 6, S4 Fig). Both EA2 and EA3 share the same molecular formula ($C_{30}H_{52}O_3$) and an identical molecular weight of 460.73 g/mol, whereas atorvastatin possesses a larger and more complex molecular formula ($C_{33}H_{35}FN_2O_5$), with a molecular weight of 558.64 g/mol (Table 6). All three compounds showed one Lipinski rule violation each, indicating acceptable oral drug-likeness. The bioavailability scores of the compounds were also comparable—0.55 for EA2 and EA3 and 0.56 for atorvastatin. The EA analogs demonstrated a lower total polar surface area (TPSA = 60.69 Å²) with 3 hydrogen bond acceptors and 3 hydrogen bond donors, whereas atorvastatin showed a considerably higher TPSA (111.79 Å²) with 6 acceptors and 4 donors. This reduced polarity of EA2 and EA3 suggests better membrane permeability and absorption potential compared with atorvastatin. In terms of lipophilicity, EA2 and EA3 exhibited higher consensus Log P values (5.58) than atorvastatin (4.94), indicating greater hydrophobic character, which may enhance passive diffusion across lipid membranes.

Pharmacokinetic assessment revealed high gastrointestinal absorption for EA2 and EA3, whereas atorvastatin showed low absorption. None of the compounds were predicted to inhibit CYP1A2 or CYP2C9, although atorvastatin was predicted to inhibit CYP2C19. With respect to dermal absorption, the EA analogs demonstrated a more favorable skin permeability profile (Log Kp = −4.21 cm/s) compared with atorvastatin (−6.19 cm/s). Water solubility prediction using the ESOL model indicated poor solubility for EA2 and EA3 (Log S = −7.04), whereas atorvastatin was moderately soluble (Log S = −5.99). From a medicinal chemistry perspective, none of the compounds triggered PAINS alerts, confirming the absence of undesirable substructures. The synthetic accessibility score for EA2 and EA3 was slightly higher (5.42) than that of atorvastatin (4.95), indicating that the EA analogs may be marginally more challenging to synthesize. Toxicological profiling predicted that EA2 and EA3 were non-toxic with respect to acute oral, dermal, and inhalation exposure, and they did not induce eye irritation or skin sensitization. However, both analogs showed potential for skin irritation and corrosion. In contrast, atorvastatin was predicted to cause eye irritation but not skin-related toxicity. Collectively, these findings suggest that the EA stereoisomers exhibit favorable ADMET characteristics comparable to or exceeding those of atorvastatin, thereby supporting their potential as promising drug candidates for further development.

**3.5.4. MM/GBSA free binding energy calculation.** The total binding free energy (ΔG Bind) was found to be −59.8 kcal/mol for the EA2–protein complex and −59.7 kcal/mol for the EA3-protein complex, both considerably lower than that

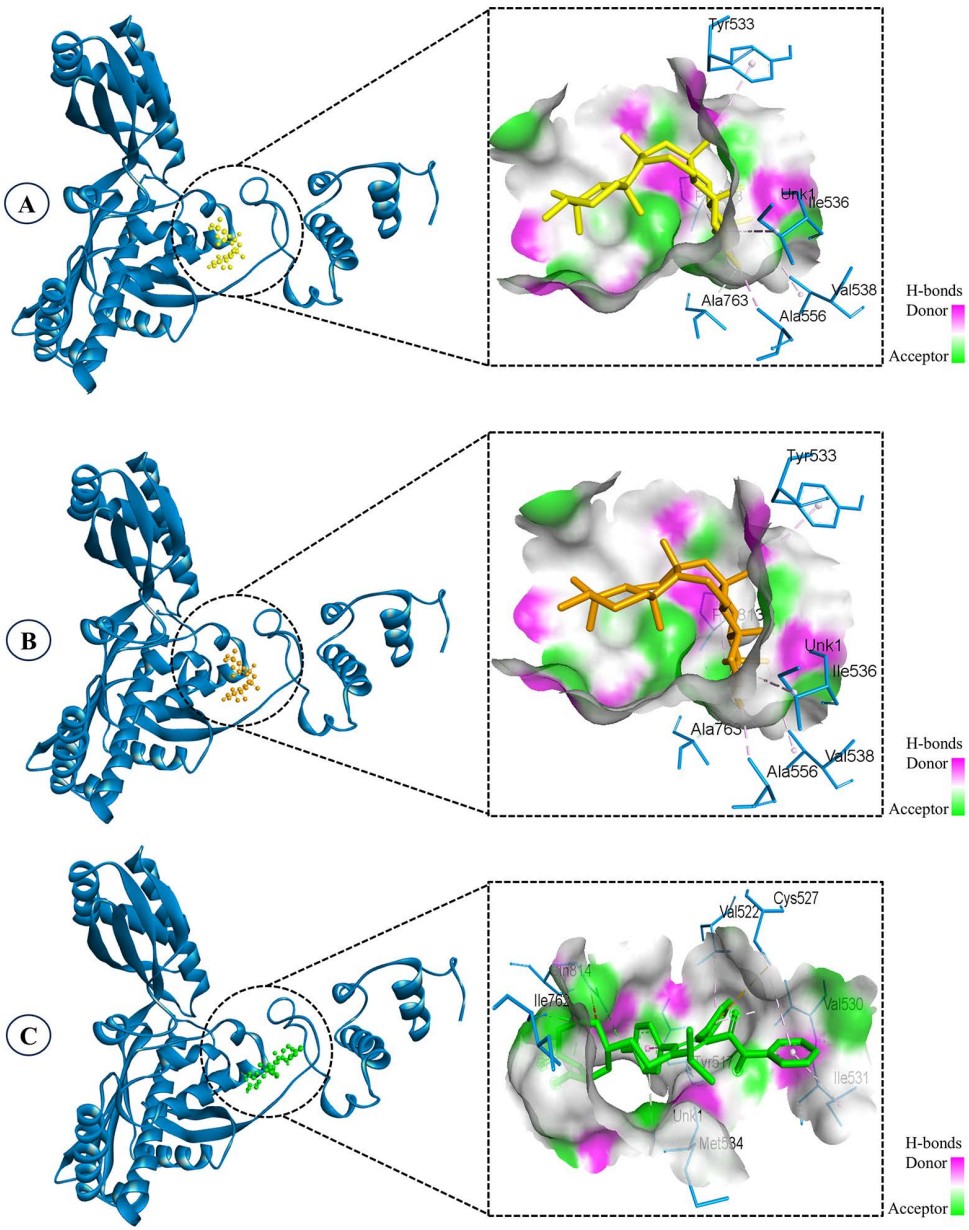

**Fig 4. Structural analogs in the binding pocket of the target protein illustrating 3D interactions. A.** EA2, **B.** EA3, **C.** Atorvastatin (control).

of the atorvastatin–protein complex (−44.7 kcal/mol) (S3 Table). This indicates that the EA analogs bind more strongly to the target protein than the control drug. Among the energy components, van der Waals interactions contributed the most to the binding, with values of −47.7 kcal/mol for both EA2 and EA3, compared to −46.0 kcal/mol for atorvastatin. Lipophilic interactions were favorable in all three complexes, with atorvastatin showing the strongest contribution (−27.9 kcal/mol), while EA2 and EA3 both contributed −20.9 kcal/mol. Electrostatic Coulomb interactions were markedly more favorable in the EA analogs (−13.9 kcal/mol each) than in atorvastatin, which showed a positive value (3.6 kcal/mol), indicating less favorable electrostatic stabilization. The covalent energy contributions were relatively small for EA2 and

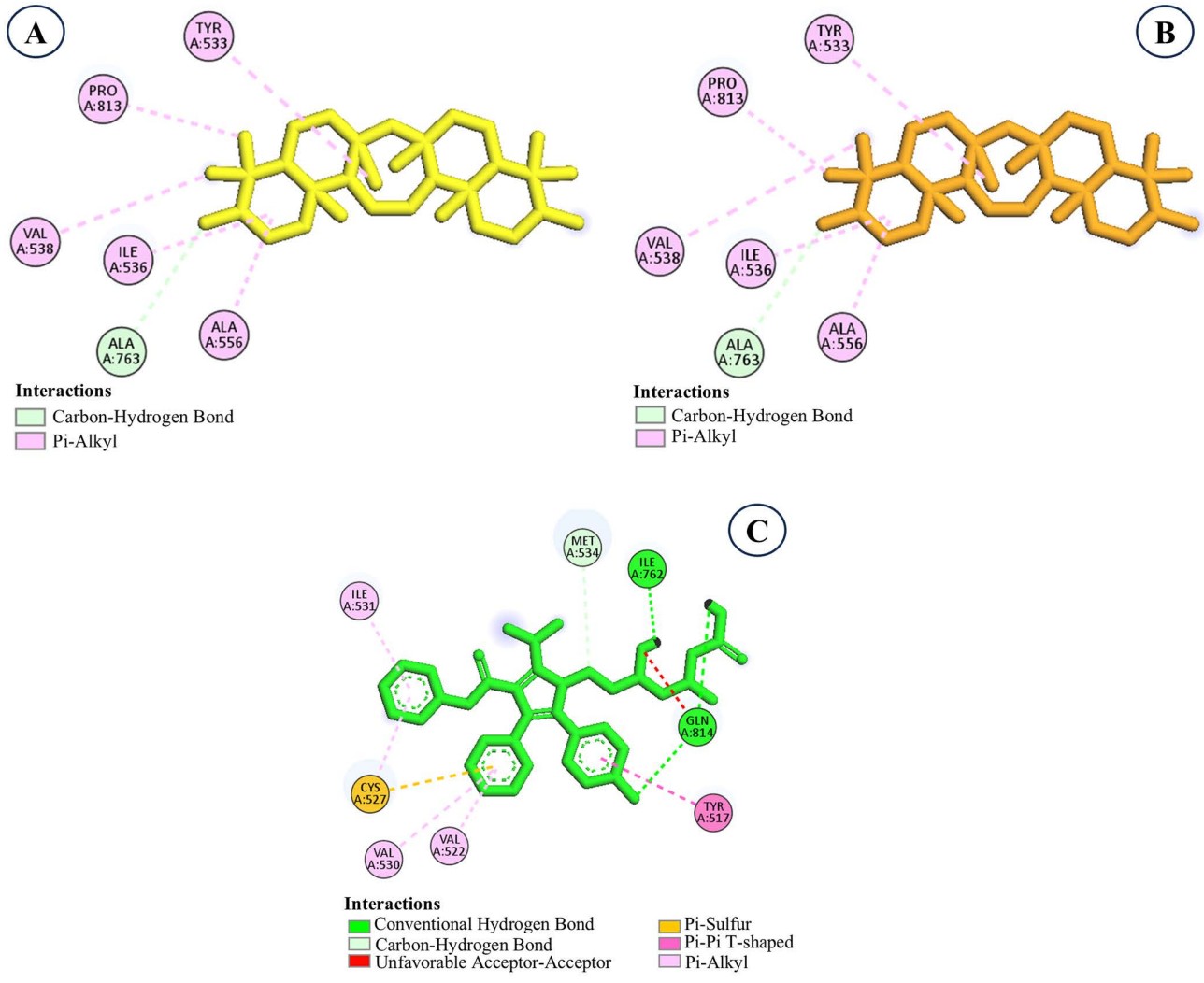

**Fig 5. Molecular interaction analysis of the structural analogs of epifriedelanol. A. EA2, B. EA3, C. Atorvastatin (control).**

EA3 (1.4 kcal/mol), but notably higher in atorvastatin (5.1 kcal/mol). Hydrogen bonding played a minimal role, contributing −1.1 kcal/mol for EA2 and EA3, and −0.9 kcal/mol for atorvastatin. Solvation energy (ΔG Solv GB) exerted a destabilizing effect across all complexes, with comparable positive values ranging from 22.5 to 22.9 kcal/mol. Overall, the MM/GBSA analysis demonstrates that both EA2 and EA3 form more stable and stronger binding to the target protein compared to atorvastatin, largely due to enhanced van der Waals and electrostatic interactions, thereby reinforcing their promise as superior lead candidates.

## 3.6. Molecular dynamics simulation

Epifriedelanol and its two structural analogs, EA2 and EA3 were considered as the final lead compounds and subjected to 500 ns MD simulations to assess their stability and structural integrity in comparison with the control, atorvastatin. For the epifriedelanol-protein complex, the root mean square deviation (RMSD) profile of the C-alpha backbone remained stable

**Table 6. ADMET properties evaluation of the selected analogs (stereoisomers) and control drug atorvastatin.**

| Parameters | Molecule | EA2 | EA3 | Atorvastatin |
|---|---|---|---|---|
| Physicochemical properties | Formula | $C_{30}H_{52}O_3$ | $C_{30}H_{52}O_3$ | $C_{33}H_{35}FN_2O_5$ |
| | Molecular weight (g/mol) | 460.73 | 460.73 | 558.64 |
| | H-bond acceptors | 3 | 3 | 6 |
| | H-bond donors | 3 | 3 | 4 |
| | Molar refractivity | 137.98 | 137.98 | 158.26 |
| | TPSA | 60.69 | 60.69 | 111.79 |
| Lipophilicity | iLOGP | 4.16 | 4.16 | 3.58 |
| | XLOGP3 | 6.90 | 6.90 | 4.96 |
| | WLOGP | 6.33 | 6.33 | 6.54 |
| | MLOGP | 5.26 | 5.26 | 3.48 |
| | Silicos-IT Log P | 5.26 | 5.26 | 6.15 |
| | Consensus Log P | 5.58 | 5.58 | 4.94 |
| Pharmacokinetics | GI absorption | High | High | Low |
| | CYP1A2 | No | No | No |
| | CYP2C19 | No | No | Yes |
| | CYP2C9 | No | No | No |
| | Log Kp (cm/s) | −4.21 | −4.21 | −6.19 |
| Water solubility (ESOL) | Log S | −7.04 | −7.04 | −5.99 |
| | Solubility (mg/ml) | 4.17E−05 | 4.17E−05 | 5.78E−04 |
| | Solubility (mol/l) | 9.05E−08 | 9.05E−08 | 1.03E−06 |
| | Class | Poorly soluble | Poorly soluble | Moderately soluble |
| Drug likeness | Lipinski (violations) | 1 | 1 | 1 |
| | Bioavailability score | 0.55 | 0.55 | 0.56 |
| Medicinal chemistry | PAINS (alerts) | 0 | 0 | 0 |
| | Synthetic accessibility | 5.42 | 5.42 | 4.95 |
| Toxicity | Acute inhalation | No | No | No |
| | Skin irritation and corrosion | Yes | Yes | No |
| | Eye irritation and corrosion | No | No | Yes |
| | Acute dermal | No | No | No |
| | Skin sensitization | No | No | No |
| | Acute oral | No | No | No |

*EA: Epifriedelanol Analogs

throughout the 500 ns trajectory without abrupt fluctuation (Fig 6A). The ligand RMSD intersected with the C-alpha backbone RMSD between 200 and 300 ns, after which the ligand maintained a steady conformation until 500 ns. The average fluctuation of the C-alpha backbone atoms remained below the accepted threshold of 3.00 Å, indicating the dynamic stability of the epifriedelanol-HMG Co-A reductase complex (Fig 6A).

The EA2-protein complex exhibited considerable fluctuations throughout the simulation, particularly during the initial 300 ns, followed by comparatively moderate deviations up to 500 ns (Fig 6B). The overall RMSD values were higher than those of the other complexes, indicating relatively lower structural stability. The ligand RMSD profile, however, showed no abrupt deviations, suggesting that the ligand remained bound within the binding pocket during the simulation.

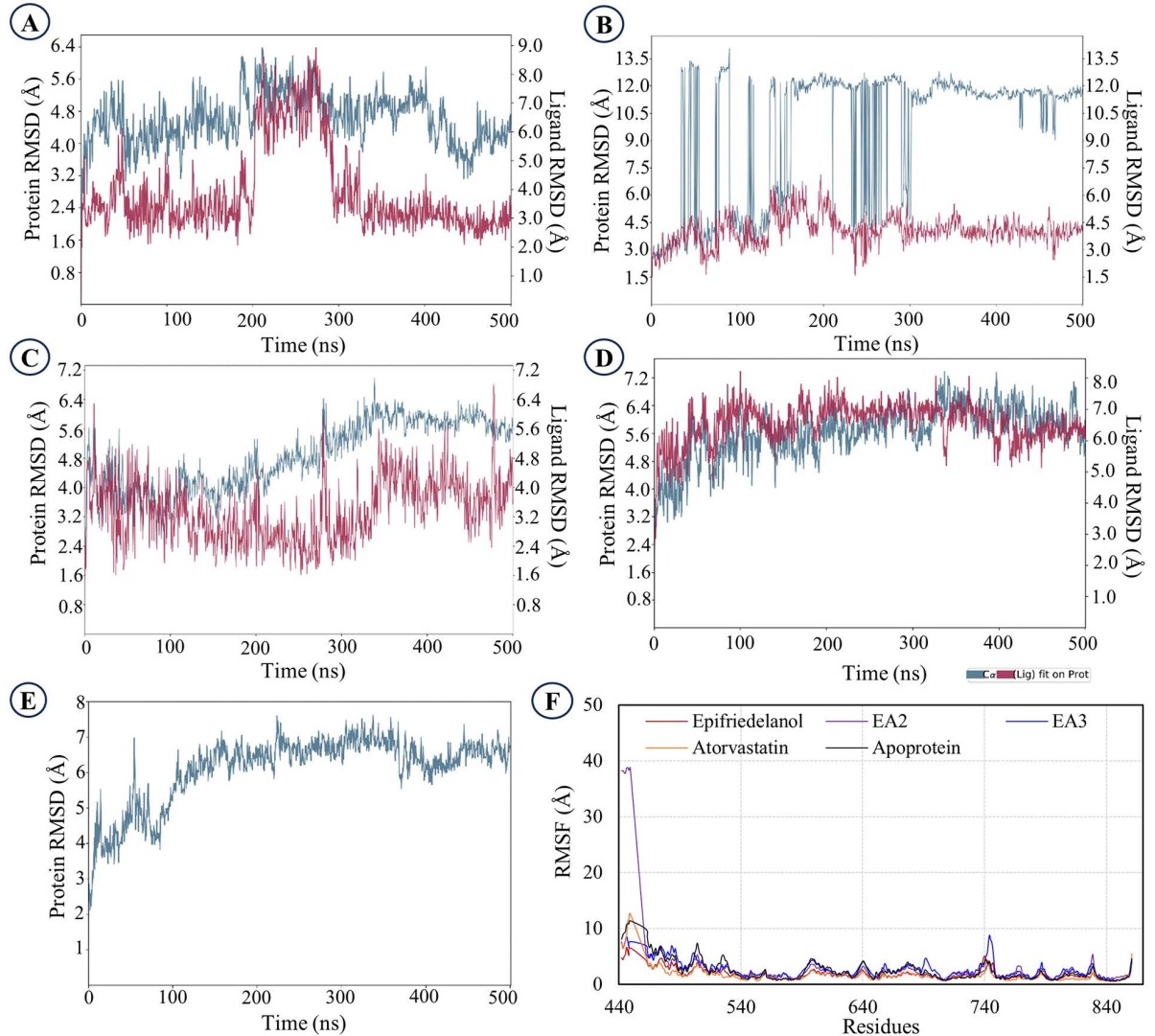

**Fig 6. RMSD and RMSF analyses of protein-ligand complexes and apoprotein. A.** RMSD of epifriedelanol-protein complex, **B.** RMSD of EA2-protein complex, **C.** RMSD of EA3-protein complex, **D.** RMSD of atorvastatin-protein complex, **E.** RMSD of the apoprotein, **F.** RMSF of the selected protein-ligand complexes and apoprotein.

The EA3-protein complex exhibited stable dynamics throughout the 500 ns simulation, as revealed by the RMSD analysis. From 0 to ~ 250 ns, the C-alpha backbone RMSD remained steady, indicating the structural stability of the protein-ligand complex. A gradual increase was observed between 250 and 300 ns, suggesting minor conformational adjustments, which subsequently stabilized near 325 ns and persisted consistently until the end of the 500 ns simulation (Fig 6C). The ligand RMSD remained low across the trajectory and closely followed the backbone fluctuations, indicating a stable binding pose of EA3 within the active site. In contrast, the atorvastatin-protein complex maintained a uniformly steady C-alpha RMSD profile throughout the 500 ns trajectory without significant deviations, further confirming the stable and persistent binding of atorvastatin (Fig 6D). The apoprotein exhibited minor movements during the initial ~150 ns, followed by a stable trajectory for the remainder of the 500 ns simulation, with no notable fluctuations (Fig 6E).

The root mean square fluctuation (RMSF) analysis was conducted to assess the flexibility of the key binding site residues upon ligand binding. In the epifriedelanol-protein complex, the RMSF values of the key residues ranged from 1.23 Å to 2.26 Å. The highest fluctuation was observed at Cys527 (2.26 Å), followed by Pro813 (1.59 Å) and Val538 (1.54 Å), while Ala763 (1.23 Å) exhibited the lowest fluctuations (Fig 6F). The average RMSF of 1.54 Å indicates relatively stable residue dynamics upon epifriedelanol binding. In comparison, the EA2-protein complex exhibited slightly higher residue flexibility, with RMSF values ranging from 1.78 Å to 2.73 Å. Pro813 showed the highest fluctuation (2.73 Å), followed by Tyr533 (2.18 Å) and Ile536 (2.13 Å), while Ala556 (1.78 Å) and Ala763 (1.84 Å) remained comparatively stable. The average RMSF of 2.11 Å for the EA2 complex suggests increased local mobility at the binding site compared with that of epifriedelanol.

The RMSF analysis revealed distinct flexibility patterns for the EA3 and atorvastatin complexes. In the EA3-protein complex, key binding site residues exhibited moderate to high fluctuations, with Pro813 showing the highest deviation (2.54 Å), followed by Tyr533 (2.41 Å), Ile536 (2.19 Å), and Val538 (2.10 Å). Ala556 (1.92 Å) and Ala763 (1.88 Å) displayed comparatively lower fluctuations. The average RMSF of 2.17 Å indicates a dynamic binding environment, likely accommodating the stereochemical configuration of EA3. In contrast, the atorvastatin-protein complex showed lower overall residue flexibility, with RMSF ranging from 0.79 Å (Ile762) to 1.78 Å (Tyr517). Residues such as Cys527, Val530, and Ile531 remained relatively stable, each fluctuating around 1.31–1.35 Å. The average RMSF of 1.35 Å for atorvastatin indicates a stronger and more stable binding in the active site compared with EA3 (Fig 6F). These findings highlight that EA3 induces greater conformational flexibility in the binding environment, possibly due to its structural features as a stereoisomer. The RMSF profile of the apoprotein revealed the flexibility of key binding-site residues, including Tyr517, Val522, Cys527, Val530, Ile531, Tyr533, Met534, Ile536, Val538, Ala556, Ile762, Ala763, Pro813, and Gln814, with corresponding fluctuations of 2.19 Å, 3.79 Å, 4.30 Å, 2.93 Å, 2.07 Å, 2.25 Å, 2.56 Å, 1.91 Å, 1.70 Å, 1.46 Å, 1.46 Å, 1.49 Å, 2.07 Å, and 1.78 Å, respectively. These fluctuation values were generally comparable to those observed in the ligand-bound complexes (Fig 6F).

The ligand property analysis, based on the mean values from the 500 ns simulation trajectory, provided detailed insights into the behavior of epifriedelanol and its analog EA2 (S5 and S6 Figs). The ligand-relative RMSD mean values indicated that all the lead compounds maintained stable conformations within the binding pocket during the simulation. EA3 (3.37 ± 0.86 Å) showed the lowest mean RMSD, suggesting the most consistent binding pose among the tested ligands. Epifriedelanol (3.91 ± 1.50 Å) and EA2 (4.05 ± 0.82 Å) also demonstrated acceptable fluctuations, reflecting moderate stability within the active site. In contrast, the atorvastatin-protein complex exhibited a notably higher ligand RMSD (6.55 ± 0.66 Å), indicating a greater positional deviation of the control ligand compared with the lead compounds.

The radius of gyration (Rg) analysis revealed that all protein-ligand complexes maintained consistent structural compactness throughout the simulation, with no major unfolding events. Among the studied compounds, epifriedelanol (4.29 ± 0.03 Å), EA2 (4.21 ± 0.05 Å), and EA3 (4.18 ± 0.05 Å) displayed closely similar and stable Rg values, indicative of compact and well-maintained protein structures upon ligand binding. In contrast, the atorvastatin-protein complex showed a noticeably higher mean Rg value (4.99 ± 0.08 Å), suggesting a relatively less compact conformation compared to the test compounds (S5 and S6 Figs).

The molecular surface area (MolSA) values of all protein-ligand complexes remained steady throughout the simulation with low standard deviations, indicating stable molecular surface exposure. The mean MolSA values for epifriedelanol, EA2, and EA3 were 379.64 ± 1.75 Å², 388.18 ± 2.19 Å², and 387.50 ± 2.18 Å², respectively. These values were considerably lower than that of atorvastatin (490.85 ± 7.46 Å²), suggesting that the lead compounds formed more compact complexes with less exposed surface area. The mean solvent accessible surface area (SASA) values for epifriedelanol, EA2, and EA3 were 116.53 ± 19.41 Å², 124.38 ± 25.81 Å², and 139.04 ± 25.25 Å², respectively. These were lower than the value observed for atorvastatin (154.33 ± 24.26 Å²). The relatively smaller SASA values of the test compounds indicate reduced solvent exposure and more compact binding conformations during the simulation. The mean polar surface area (PSA)

values were 33.53 ± 1.65 Å² for epifriedelanol, 108.58 ± 2.30 Å² for EA2, 108.68 ± 2.34 Å² for EA3, and 172.84 ± 7.92 Å² for atorvastatin. The lower PSA values of the lead compounds suggest moderate polarity and potentially improved membrane permeability compared with the control drug, which showed a higher polar surface area (S5 and S6 Figs).

The protein-ligand contact analysis revealed distinct interaction profiles for epifriedelanol, EA2, EA3, and the reference drug atorvastatin (Fig 7). In the epifriedelanol-protein complex, Ile536 showed the highest interaction fraction, characterized primarily by dominant hydrogen bonds, supplemented by occasional water-bridge contacts and transient hydrophobic interactions. Notably, no ionic contacts were detected throughout the trajectory.

For both EA2 and EA3, which are stereoisomers, Pro813 consistently exhibited the highest interaction fraction. In both complexes, Pro813 engaged in stable hydrogen bonding as the predominant interaction type, accompanied by water-bridge and hydrophobic contacts, indicating a robust and nearly identical binding pattern likely attributable to their shared stereochemistry (Fig 7). No ionic interactions were detected in either case. In contrast, the atorvastatin-protein complex displayed a distinct interaction profile, with Gln814 showing the highest interaction fraction. Water-bridge contacts were found to be dominant, supported by hydrogen bonds. These findings suggest that EA2 and EA3 form stable and similar interactions with the protein, particularly with Pro813, while epifriedelanol preferentially engages Ile536 and atorvastatin adopts a distinct interaction orientation centered on Gln814.

Hydrogen-bond occupancy and protein-ligand center-of-mass (PL-COM) analyses were carried out to evaluate the stability and persistence of the ligand-protein interactions during the 500 ns molecular dynamics simulation (Fig 8). The total number of hydrogen bonds formed throughout the trajectory followed the order atorvastatin > EA3 > EA2 > epifriedelanol, with 1425, 940, 896, and 714 cumulative interactions, respectively. The control ligand, atorvastatin, maintained the highest number of hydrogen bonds, indicating strong and stable electrostatic interactions with the active site residues. Among the natural analogs, EA3 exhibited the highest hydrogen-bond count, suggesting relatively stable engagement with the binding pocket compared to EA2 and epifriedelanol (Figs 8A–D).

The PL-COM analysis demonstrated that all complexes maintained mean separations within the 1.6–2.0 Å range, indicating that none of the ligands experienced major displacement from the binding pocket during the simulation. The mean PL-COM distances were 1.63 ± 0.44 Å for EA2, 1.74 ± 0.53 Å for EA3, 1.80 ± 0.53 Å for epifriedelanol, and 2.05 ± 0.36 Å for atorvastatin. Among the tested compounds, EA2 exhibited the lowest mean distance with the least fluctuation, suggesting a more consistent binding orientation throughout the trajectory. In contrast, the atorvastatin-complex showed the highest mean PL-COM value, indicating comparatively greater flexibility of the control drug within the binding pocket (Fig 8E).

### 3.7. Post-MD simulation free binding energy evaluation

Post-molecular dynamics MM/GBSA calculations were performed to evaluate the binding affinities and energetic contributions of the protein-ligand complexes over the 500 ns simulation, which revealed suitability of the leads as potential drug candidates (Table 7). The lead compounds exhibited stronger binding to the protein compared to atorvastatin. Epifriedelanol, EA2, EA3, and atorvastatin demonstrated mean free binding energies of −39.5, −31.3, −43.7, and −21.4 kcal/mol, respectively. In the free binding energy graph, EA3 consistently maintained the lowest energy values throughout most of the simulation frames, indicating a highly stable interaction with the protein (Fig 9).

Analysis of individual energy components indicated that van der Waals and electrostatic interactions were the main stabilizing forces, emphasizing the importance of hydrophobic and electrostatic complementarity within the binding pocket. EA3 displayed the most favorable Coulombic energy (−40.5 kcal/mol) and substantial van der Waals interactions (−33.4 kcal/mol), while hydrogen-bonding (−4.8 kcal/mol) and lipophilic interactions (−5.6 kcal/mol) provided additional stabilization. Contributions from solvation and ligand strain energy opposed binding, reflecting the energetic cost of desolvation and conformational adjustments required for complex formation. Despite these penalties, all the lead compounds showed better free binding energies over the control atorvastatin.

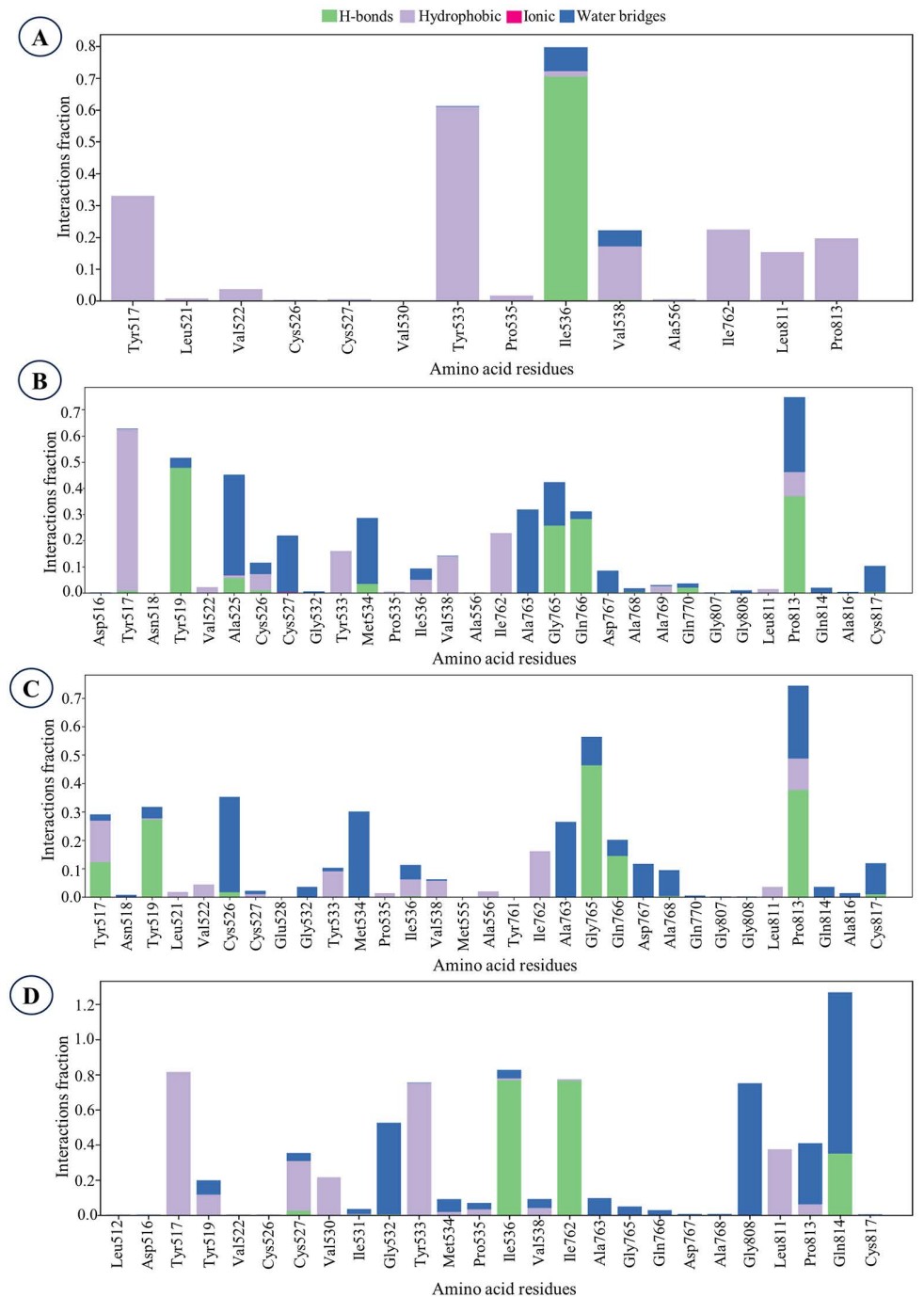

**Fig 7. Protein-ligand contacts illustrating the interaction fractions of the ligands. A.** Epifriedelanol, **B.** EA2, **C.** EA3, **D.** Atorvastatin (control).

## 3.8. PCA and Gibbs FEL study

PCA revealed the key dynamic features of the selected lead compounds in comparison with the control drug atorvastatin (Fig 10). Epifriedelanol and its two analogs exhibited phase-space distribution comparable to that of atorvastatin. Among

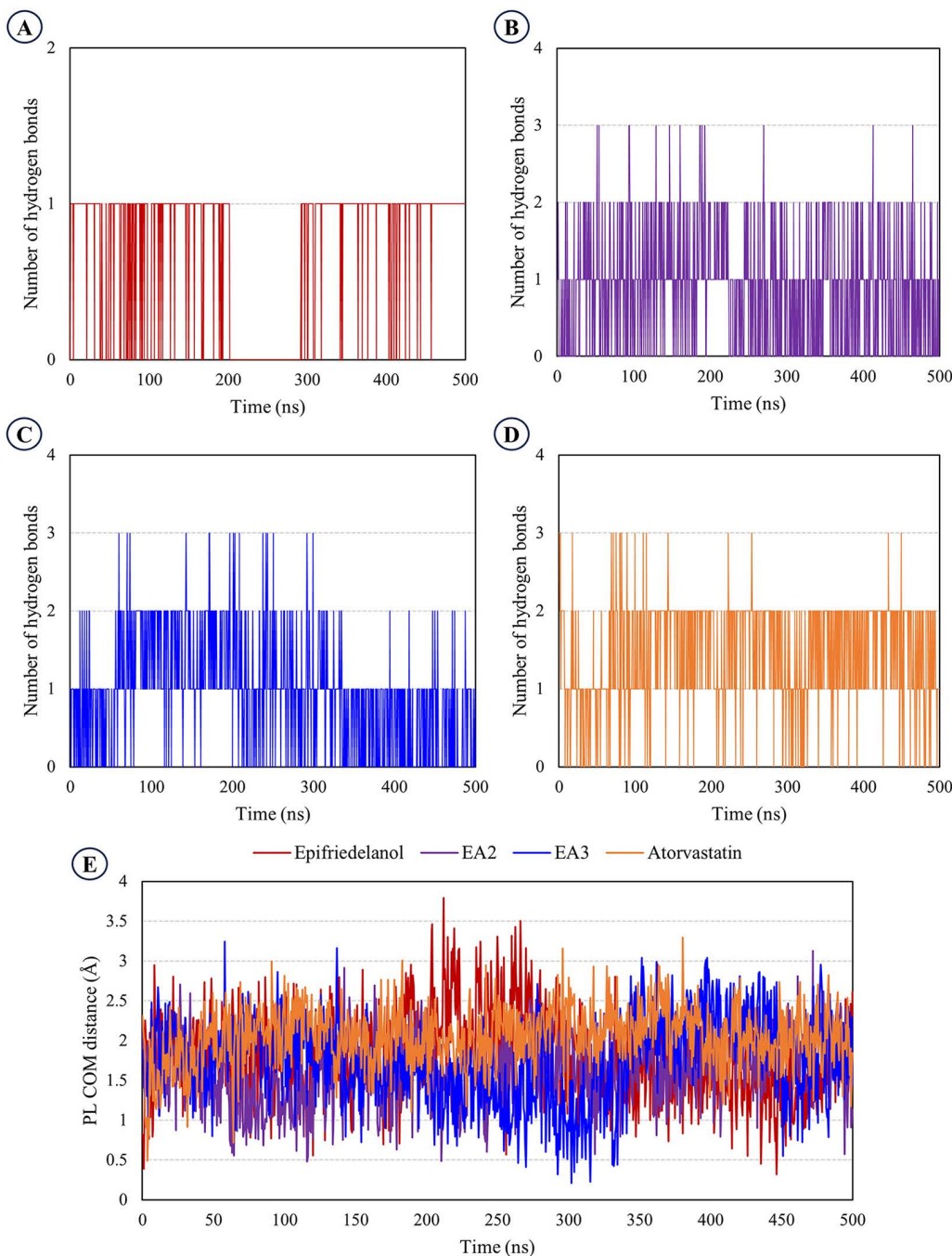

**Fig 8. Hydrogen bonds and center-of-mass (PL-COM) analyses of the lead compounds and control drug. A.** Epifriedelanol H-bond profile, **B.** EA2 H-bond profile, **C.** EA3 H-bond profile, **D.** Atorvastatin H-bond profile, **E.** Center-of-mass profiles.

the three leads, EA2 demonstrated the least phase-space, followed by EA3 and epifriedelanol. Upon superimposition, all the three lead compounds showcased a distribution pattern comparable to the control drug atorvastatin.

The Gibbs free energy landscape (FEL) corroborated the PCA results, revealing a rugged energy surface with multiple local minima distributed across the conformational space (Fig 10). Notably, none of the compounds, including the control

**Table 7. MM/GBSA calculation with mean values of all the frames after the molecular dynamics simulation.**

| Compounds | Epifriedelanol | EA2 | EA3 | Atorvastatin (control) |
|---|---|---|---|---|
| ΔG Bind (kcal/mol) | −39.5 | −31.9 | −43.7 | −21.4 |
| ΔG Coulomb (kcal/mol) | −9.7 | −13.2 | −40.5 | −3.8 |
| ΔG Covalent (kcal/mol) | 1.8 | 4.3 | 2.8 | 1.2 |
| ΔG Hbond (kcal/mol) | −3.1 | −2.6 | −4.8 | −1.3 |
| ΔG Lipo (kcal/mol) | −6.3 | −6.0 | −5.6 | −3.5 |
| ΔG Solv GB (kcal/mol) | 12.3 | 20.6 | 39.1 | 4.3 |
| ΔG vdW (kcal/mol) | −32.9 | −34.5 | −33.4 | −17.7 |
| Ligand strain energy (kcal/mol) | 8.1 | 11.9 | 11.0 | 4.5 |

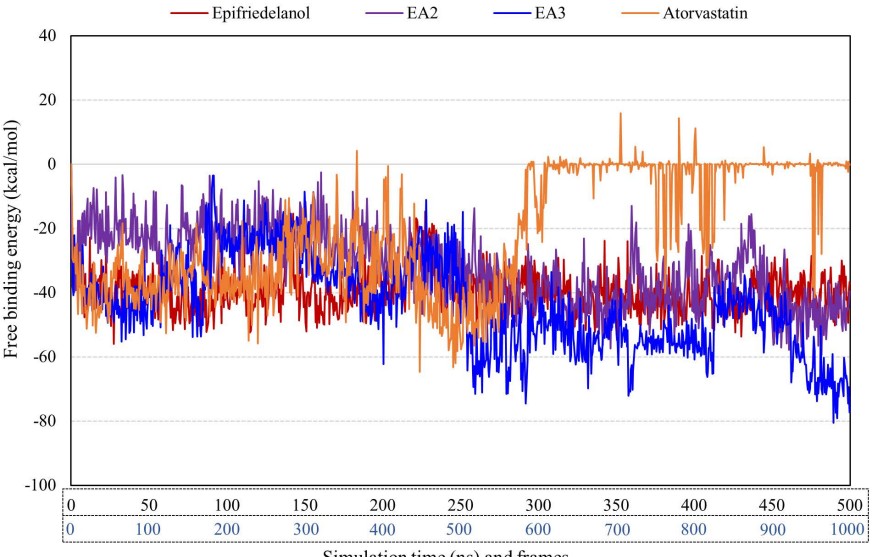

**Fig 9. MM/GBSA free binding energy analysis (per frame) after molecular dynamics simulation.**

atorvastatin, showed a single dominant energy minimum. This suggests that the systems remained dynamically flexible throughout the simulation, sampling diverse conformations without converging to a highly stable energetic state. Such a distribution supports the conformational adaptability of the lead compounds within the biological environment.

### 3.9. Evaluation of the drug target class

The potential drug target classes of epifriedelanol, its two analogs (EA2 and EA3), and the reference drug atorvastatin are illustrated in Fig 11. Epifriedelanol, EA2, and EA3 predominantly targeted nuclear receptors, each with a predicted probability of 26.7%, indicating a strong potential for interaction with members of this protein family. This convergence suggests a shared mechanism of action and pharmacological profiles among the three lead compounds.

In contrast, atorvastatin displayed a broader target class distribution, with equal probabilities (20%) for three distinct classes: Family A G-protein-coupled receptors (GPCRs), nuclear receptors, and erasers. The presence of "erasers" as a predicted target class indicates potential interactions with epigenetic regulatory proteins, such as histone deacetylases, which influence chromatin remodeling and gene expression regulation. These findings suggest that while epifriedelanol

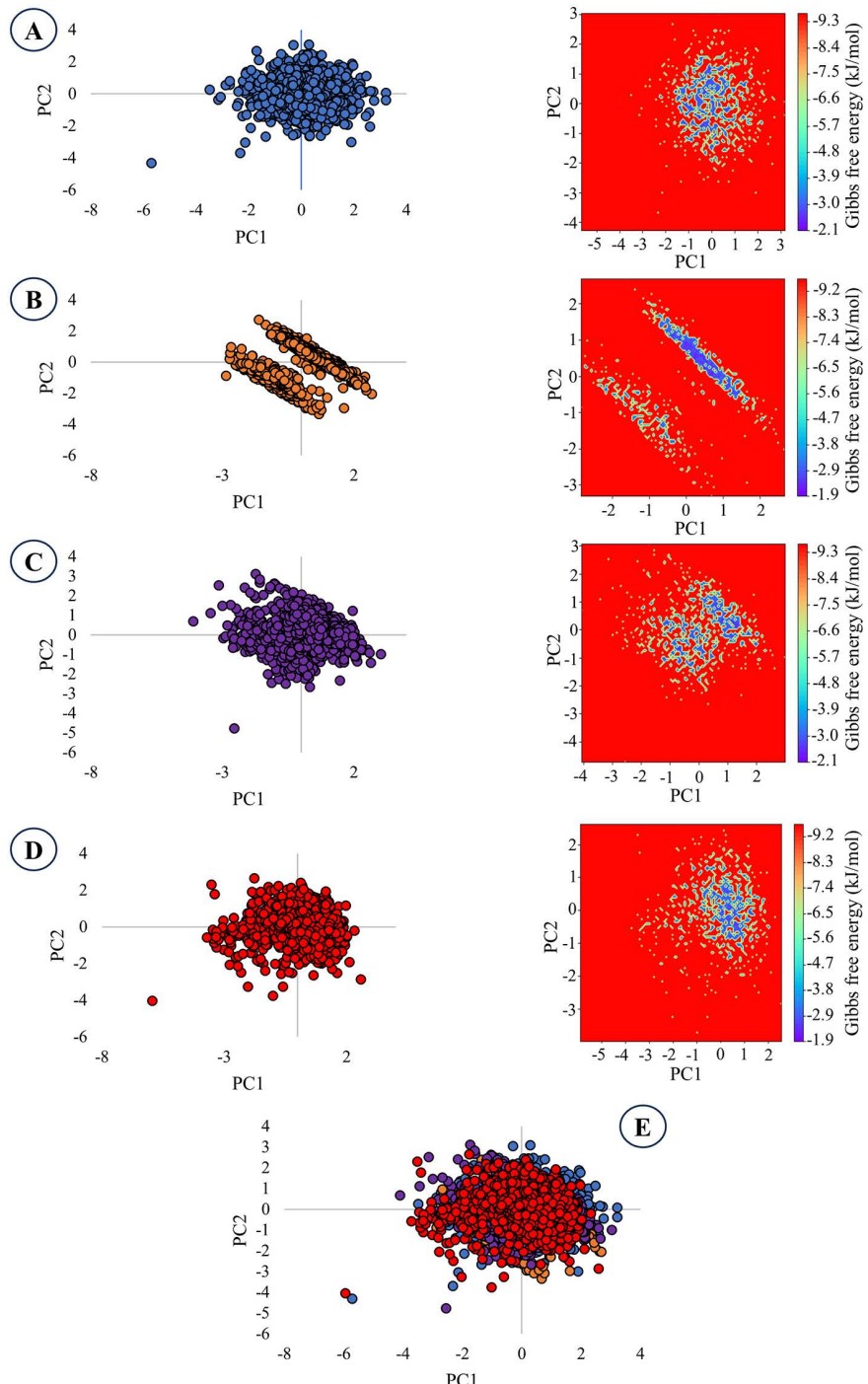

**Fig 10. PCA and Gibbs FEL observations of the selected leads and control. A.** Epifriedelanol, **B.** EA2, **C.** EA3, **D.** Atorvastatin, **E.** Superimposition of the three leads and atorvastatin.

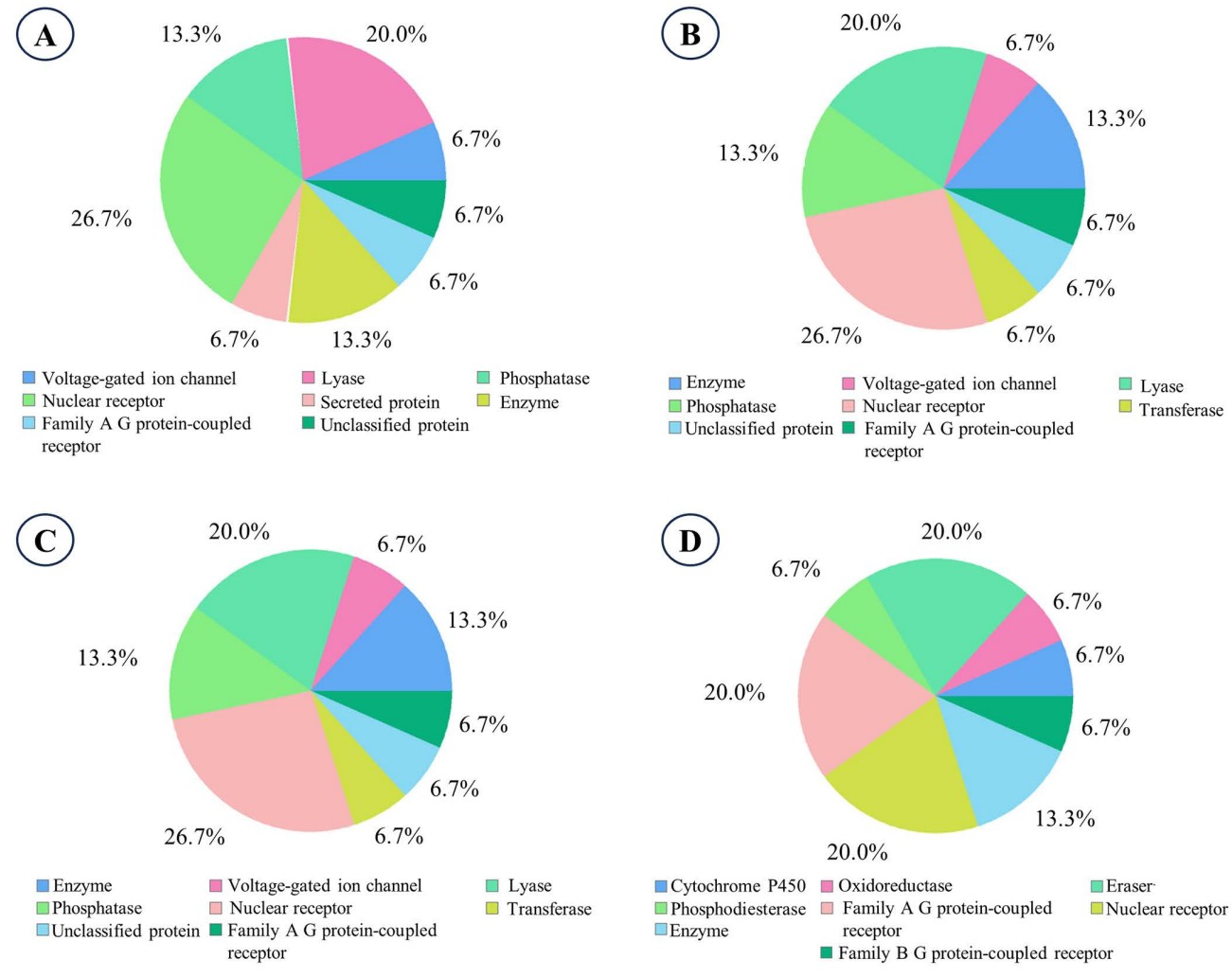

**Fig 11. Drug target class evaluation of the lead compounds and control. A.** Epifriedelanol, **B.** EA2, **C.** EA3, **D.** Atorvastatin (control).

and its analogs are predicted to act primarily through nuclear receptor modulation, atorvastatin may exert its effect via multiple target classes, including epigenetic and receptor-mediated pathways.

## 4. Discussion

In this study, the catalytic domain of human HMG-CoA reductase (PDB ID: 1HWK) was selected as the receptor for the molecular docking of *M. cordata* phytocompounds, consistent with previous studies [53–55]. The 1HWK crystal structure represents the enzyme in complex with atorvastatin at 2.22 Å resolution, thus providing an experimentally validated active site for ligand recognition and benchmarking. This protein has been repeatedly used in recent phytochemical-based molecular docking studies, such as with *Triphala* constituents [54], *Ficus religiosa* fruit extracts [55], and statin derivative drug-design [53], demonstrating its reliability in evaluating HMG-CoA reductase inhibition. Although it encompasses only the catalytic domain, this region is sufficient for inhibitor binding, as confirmed by structural and mechanistic analyses. The presence of co-crystallized atorvastatin further defines the key binding interactions, enabling accurate grid placement and comparative assessment of the phytocompound poses. Thus, employing the 1HWK structure ensures methodological

consistency with prior studies [53–55], while providing a robust and widely accepted template for the virtual screening of HMG-CoA reductase inhibitors.

It is worth noting that a more recent Cryo-EM structure of the catalytic domain of human HMG-CoA reductase (PDB ID: 8S6B, released on February 2025) is available at 2.06 Å resolution [56]. However, this structure represents the apo form of the enzyme, lacking a bound ligand. In contrast, the 1HWK structure contained the co-crystallized inhibitor atorvastatin within the catalytic pocket, allowing accurate definition of the binding site and providing experimentally validated ligand-protein interactions. Therefore, 1HWK remains a more appropriate choice for structure-based docking and comparative binding analysis, despite the slightly higher resolution of the 8S6B structure. In the present study, redocking of the co-crystallized ligand into 1HWK yielded an RMSD of 2.163 Å between the predicted and experimental poses (Fig. 1), which is marginally above the commonly cited 2.0 Å benchmark for docking validation. However, RMSD should not be interpreted as a strict cutoff; rather, values within the 2.0–3.0 Å range have been reported as acceptable, particularly when the correct ligand orientation and key binding interactions are preserved [57,58]. This marginal increase in RMSD between the predicted and experimental poses is consistent with previously reported docking studies and validates the docking protocol employed in the present investigation.

In comparison with the study of Khori et al. [59], where monacolin k combined with B12N12 nanoparticles achieved docking scores of −9.8 and −8.9 kcal/mol, higher than monacolin k alone (−7.3 kcal/mol) and rosuvastatin (−8.8 kcal/mol), the present findings with *M. cordata* phytocompounds show a similar trend. Epifriedelanol analogs (EA2 and EA3) exhibited docking scores of −9.3 kcal/mol, while the parent compound epifriedelanol scored −8.6 kcal/mol, both outperforming atorvastatin (−7.7 kcal/mol). These results emphasize that structural modification, whether via nanoparticle functionalization or analog design, can substantially enhance the inhibitory potential against HMG-CoA reductase compared with standard statins [59]. The importance of structural analogs in optimizing drug-target interactions is further demonstrated by Reyaz et al. [60], who designed hydroxychloroquine analogs to occupy key water-mediated positions in the SARS-CoV-2 main protease, thereby strengthening the binding and potential inhibitory effects. Analog construction enables subtle structural modifications around a parent compound to refine interactions with critical residues and structural features of a target protein. In a similar manner, epifriedelanol from *M. cordata* was first identified as a promising HMG-CoA reductase inhibitor, and the subsequent generation of 451 analogs led to the identification of EA2 and EA3, which showed superior binding affinity and favorable pharmacokinetic properties. Both studies highlight the power of analog-based design as a systematic strategy to explore the chemical space, refine molecular interactions, and identify novel compounds with enhanced potency, selectivity, and drug-likeness for developing optimized therapeutic candidates [61,62].

Although the canonical catalytic pocket of HMG-CoA reductase is well-established as the primary statin-binding site, recent studies have revealed the presence of additional druggable cavities and regulatory interfaces within the enzyme. In the present study, the CASTp server was employed to systematically explore all possible surface-accessible pockets, allowing the identification of a prominent cavity encompassing residues such as Tyr517, Ile531, Met534, and Pro813. Interestingly, this region partially overlaps with residues (Ile531, Met534) previously reported to contribute to the dimer stability and conformational regulation of the receptor [63]. In addition, Kanu et al. [64] demonstrated that natural polyphenols such as EGCG, ECG, and EC can effectively interact with Glu528, Met534, Ile536, and Lys735 (residues distinct from the canonical statin site), yet display strong inhibitory activity toward HMG-CoA reductase. These findings collectively suggest that the CASTp-identified cavity represents a biologically meaningful and pharmacologically relevant region that may facilitate alternative or allosteric modes of inhibition. The observation that both atorvastatin and the lead compounds exhibited favorable binding within this CASTp-predicted site further reinforces the plausibility of this pocket as a potential regulatory hotspot for the small-molecule modulation of HMG-CoA reductase.

Epifriedelanol has previously been reported as a natural compound with potential antihyperlipidemic and antidiabetic activities [65,66]. Ramdhani and Mustarichie [65] demonstrated that epifriedelanol from *Guazuma ulmifolia* exhibits strong binding affinity to lanosterol 14α-demethylase (−11.2 kcal/mol) and squalene synthase (−10.3 kcal/mol), indicating its

potential to modulate cholesterol biosynthesis, whereas its interaction with NPC1L1 was weaker (−2.5 kcal/mol). Similarly, Wibowo et al. [66] showed that epifriedelanol from *Aquilaria malaccensis* leaves contributes to α-glucosidase inhibition, further supporting its metabolic regulatory potential. These findings align with the present study, which identified epifriedelanol and its analogs EA2 and EA3 as promising HMG-CoA reductase inhibitors, reinforcing the potential of epifriedelanol derivatives as versatile metabolic regulators. Notably, epifriedelanol and its analogs each showed one violation of Lipinski's rule of five, consistent with the control compound atorvastatin, which also presented a single violation. Importantly, besides this exception, all other ADMET properties of the identified leads were within acceptable ranges, suggesting favorable drug-likeness, pharmacokinetics, and safety profiles. This indicates that the observed Lipinski violation does not diminish their potential as viable drug candidates, particularly since many clinically approved drugs, including statins, share similar exceptions. The ADMET results were also consistent with other recent studies [44,49,51]. SwissADME provided detailed predictions of drug-likeness and pharmacokinetic behavior, while toxicity predictions from STopTox complemented these data, together offering a complete ADMET assessment.

In this study, 500 ns molecular dynamics simulations were performed to evaluate the structural stability and binding behavior of the three lead compounds–epifriedelanol, EA2 and EA3 with the HMG-CoA reductase protein. The post-MD analyses collectively confirmed the persistent binding and structural stability of the lead compounds within the HMG-CoA reductase active site. Although the global protein C-alpha RMSD for EA2 was elevated (9.59 Å), the RMSF analysis indicated that this was due to the local flexibility of specific binding-site residues rather than ligand dissociation. The average RMSF values were 1.54 Å for epifriedelanol, 2.11 Å for EA2, 2.17 Å for EA3, and 1.35 Å for atorvastatin, reflecting slightly increased local mobility for EA2 and EA3. Residues such as Pro813, Tyr533, and Ile536, exhibited higher fluctuations to accommodate the stereochemistry of EA2 and EA3, while key residues such as Ala556 and Ala763 remained stable, supporting a well-maintained binding environment. The ligand-centric and structural compactness metrics consistently demonstrated stable binding retention. Specifically, EA2 exhibited an acceptable ligand-relative RMSD (4.05 ± 0.82 Å) and, critically, the lowest mean protein-ligand center-of-mass (PL-COM) distance (1.63 ± 0.44 Å), confirming a persistent and close association within the pocket (Fig 8E). Consistently, all the lead compounds maintained structural compactness and solvent shielding, as reflected by lower Rg, MolSA, and SASA values compared with the control, atorvastatin (S5 and S6 Figs). Persistent hydrogen-bond interactions (cumulative counts: 896–940) further supported sustained polar contacts with active-site residues (Figs 8A–D). Together, these complementary data indicate that the apparent C-alpha RMSD elevation and moderate local RMSF fluctuations for EA2 and EA3 result from normal local conformational adaptation and protein flexibility, rather than complex destabilization, thus validating that all lead compounds remained stable and compactly bound throughout the 500 ns simulation.

While most drug design studies employ 100 ns simulations, such a timescale may not adequately capture slower conformational transitions, rare binding events, or long-timescale fluctuations of protein-ligand complexes [67,68]. Extending the simulations to 500 ns enable a more comprehensive assessment of the dynamic behavior, improving confidence in the stability and persistence of the molecular interactions. This longer simulation timescale also enhances the robustness of the binding free energy calculations, essential dynamics, and free energy landscape profiling, thereby strengthening the predictive power of the computational framework. By employing extended simulations to 500 ns, the present study provides a more rigorous methodological approach for evaluating natural product-derived and analog-based inhibitors, ensuring that the identified leads exhibit sustained interactions and structural stability under physiologically relevant timescales [51,69].

In the present investigation, binding free energy calculations were performed both immediately after docking and following molecular dynamics simulations to assess the stability and affinity of the selected ligands. Post-docking MM/GBSA calculations provided a rapid and quantitative estimate of ligand-protein interactions, enabling the efficient prioritization of potential leads. This approach allowed the identification of epifriedelanol as the most promising phytocompound from *M. cordata*, while other candidates such as taraxasterol were excluded despite favorable ADMET profiles (S2 Table). The

structural analogs EA2 and EA3 were subsequently shortlisted based on their higher predicted binding affinities compared to the control drug atorvastatin. The use of MM/GBSA at this stage is consistent with previous studies. Shimu et al. [70] employed post-docking energy calculations to screen phytochemicals against the Dengue virus NS2B/NS3 protein before molecular dynamics simulations. Ahmed et al. [71] applied a similar strategy to prioritize natural compounds from *Piper betle* against alpha-amylase and alpha-glucosidase. Ahmed et al. [51] performed MM/GBSA across all 1000 frames of a 500 ns MD simulation to robustly evaluate the binding stability of *Toxicodendron succedaneum* phytochemicals against ERK2. However, it is important to note that post-docking MM/GBSA calculations are based on static docking poses and therefore do not account for the dynamic nature of the protein and ligand, solvent effects, or time-dependent conformational fluctuations. As a result, these calculations can overestimate the binding affinity and provide limited insight into the stability of the complex under physiological conditions. To address these limitations, we performed 500 ns molecular dynamics simulations followed by MM/GBSA calculations across 1000 frames, which provided a time-averaged and more realistic assessment of the binding free energies. This combined strategy confirmed that epifriedelanol, EA2, and EA3 maintained stronger and more stable interactions than atorvastatin throughout the simulation, validating their potential as robust lead compounds. Although the post-MD binding free energy profile of atorvastatin exhibited transient positive ΔG_ bind values during the latter part of the simulation, this behavior is best interpreted as a consequence of increased conformational flexibility and weakened interaction strength rather than complete ligand dissociation. MM/GBSA calculations are inherently sensitive to instantaneous structural fluctuations and solvent exposure, particularly for flexible ligands; consequently, short-lived positive energy values may arise during periods of partial disengagement. Importantly, ligand-centric analyses confirmed that atorvastatin remained associated with the catalytic pocket throughout the 500 ns simulation, as evidenced by stable protein-ligand center-of-mass distances and the persistence of intermittent hydrogen-bond interactions (Fig 8).

SwissTargetPrediction analysis revealed nuclear receptors as the predominant target class (26.7% probability) for all three lead compounds. Nuclear receptors such as LXRs, FXR, and PPARs are key regulators of lipid and cholesterol homeostasis, and their activity is modulated by metabolites generated downstream of the mevalonate pathway governed by HMG CoA-reductase [72,73]. Thus, while our structure-based docking confirmed the direct interaction of epifriedelanol and its analogs with the HMG CoA-reductase, the target class prediction demonstrates that these compounds may also modulate lipid metabolism indirectly through nuclear receptor-mediated transcriptional regulation.

Despite the promising outcomes from molecular docking, ADMET profiling, MM/GBSA calculations, and molecular dynamics simulations, the inherent limitations of *in silico* approaches to be acknowledged. Computational predictions, while efficient for prioritizing potential candidates, cannot fully capture the complexity of biological systems, including metabolic variability, off-target interactions, and *in vivo* physiological responses [14]. Therefore, the identified leads such as epifriedelanol, EA2, and EA3 require *in vitro* validation of their HMG-CoA reductase inhibitory activity, followed by *in vivo* studies to evaluate pharmacokinetics, toxicity, and therapeutic efficacy. Future experimental work may include spectrophotometric or fluorescence-based HMG-CoA reductase enzyme inhibition assays to quantify the direct inhibitory activity, and cell-based cholesterol biosynthesis or uptake assays using HepG2 or H9c2 cells to assess the functional lipid-lowering effects. *In vivo* validation could be performed in diet-induced hypercholesterolemic rat or mouse models to evaluate serum lipid profiles, hepatic enzyme levels, and histopathological changes in cardiac and hepatic tissues. Such experimental validation is essential and recommended to confirm the translational potential of these candidates as viable cholesterol-lowering agents.

## 5. Conclusion

This study provides a comprehensive computational evaluation of *M. cordata* phytoconstituents for their potential to inhibit HMG-CoA reductase and manage hypercholesterolemia. Among the 91 screened compounds, epifriedelanol and its analogs EA2 and EA3 emerged as the most promising candidates, demonstrating strong predicted binding

affinities, favorable ADMET properties, and structural stability in extended molecular dynamics simulations. MM/GBSA and essential dynamics analyses further supported their potential efficacy, while target prediction suggested nuclear receptors as the dominant target class. These findings highlight epifriedelanol and its analogs as viable leads for natural cholesterol-lowering agents. They warrant future experimental validation, including *in vitro* HMG-CoA reductase inhibition assays to determine $IC_{50}$ values and cell-based studies in hepatocyte models to assess the effects on cholesterol synthesis and lipid uptake. Additionally, *in vivo* evaluation in diet-induced hypercholesterolemic rodent models is recommended to examine serum lipid profiles, liver function, and safety, thereby confirming the therapeutic potential and translational relevance of these leads.

## Supporting information

**S1 Fig. Identification of the top-ranked active site in the target protein.** The cavity ranked first, exhibiting the largest surface area and volume, was selected as the final binding site. A. Cavity ranked 1, B. Cavity ranked 2.
(PDF)

**S2 Fig. Two-dimensional chemical structures of the 91 phytocompounds of *M. cordata* and the control drug atorvastatin.** Numbers are arranged according to the S1 Table.
(PDF)

**S3 Fig. Bioavailability radar plots illustrating the key physicochemical properties (lipophilicity, size, polarity, insaturation, solubility, and flexibility) of the *M. cordata* phytocompounds and control drug.** A. Taraxasterol, B. Epifriedelanol, C. Friedelin, D. Atorvastatin (control).
(PDF)

**S4 Fig. Bioavailability radar plots showing the key physicochemical properties (lipophilicity, size, polarity, insaturation, solubility, and flexibility) of the epifriedelanol analogs and reference drug.** A. EA2, B. EA3, C. Atorvastatin (control).
(PDF)

**S5 Fig. Evaluation of ligand properties of the two lead candidates.** A. Epifriedelanol, B. EA2.
(PDF)

**S6 Fig. Evaluation of ligand properties of a lead compound and a control drug.** A. EA3, B. Atorvastatin (control).
(PDF)

**S1 Table. Molecular docking analysis of *M. cordata* phytocompounds and control drug elucidating binding affinity.**
(PDF)

**S2 Table. Free binding energy analysis of the top selected phytocompounds of *M. cordata*.**
(PDF)

**S3 Table. Free binding energy calculation of the selected epifriedelanol analogs and control drug.**
(PDF)

## Acknowledgments

The authors sincerely acknowledge the logistic support provided by the University of Dhaka during the conduct of this research.

## Author contributions

**Conceptualization:** Miruna Banu, Sheikh Sunzid Ahmed, M. Oliur Rahman.

**Data curation:** Miruna Banu, Sheikh Sunzid Ahmed.

**Formal analysis:** Miruna Banu, Sheikh Sunzid Ahmed, M. Oliur Rahman.

**Investigation:** Miruna Banu, Sheikh Sunzid Ahmed, M. Oliur Rahman.

**Methodology:** Miruna Banu, Sheikh Sunzid Ahmed.

**Project administration:** M. Oliur Rahman.

**Resources:** Sheikh Sunzid Ahmed, Momtaz Begum, M. Oliur Rahman.

**Supervision:** M. Oliur Rahman.

**Validation:** Momtaz Begum, M. Oliur Rahman.

**Visualization:** Sheikh Sunzid Ahmed, Momtaz Begum, M. Oliur Rahman.

**Writing – original draft:** Miruna Banu, Sheikh Sunzid Ahmed.

**Writing – review & editing:** Momtaz Begum, M. Oliur Rahman.

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
