## [Decision Letter · Decision Letter 0]

22 Oct 2025

Dear Dr. Rahman,

Thank you for submitting your manuscript to PLOS ONE. After careful consideration, we feel that it has merit but does not fully meet PLOS ONE’s publication criteria as it currently stands. Therefore, we invite you to submit a revised version of the manuscript that addresses the points raised during the review process.

We look forward to receiving your revised manuscript.

Kind regards,

Chandrabose Selvaraj, Ph.D.

Academic Editor

PLOS ONE

Journal Requirements:

3. We note that your Data Availability Statement is currently as follows: “All relevant data are available within the manuscript”

4. ***If the reviewer comments include a recommendation to cite specific previously published works, please review and evaluate these publications to determine whether they are relevant and should be cited. There is no requirement to cite these works unless the editor has indicated otherwise. ***

Additional Editor Comments :

If reviewers recommend citations that are not relevant to the manuscript, authors are encouraged to decline them. The journal assures that choosing not to include such irrelevant citations will not affect the editorial decision on the manuscript.

Reviewers' comments:

Reviewer's Responses to Questions

**Comments to the Author**

1. Is the manuscript technically sound, and do the data support the conclusions?

Reviewer #1: Yes

Reviewer #2: Yes

Reviewer #3: Yes

2. Has the statistical analysis been performed appropriately and rigorously?

Reviewer #1: Yes

Reviewer #2: Yes

Reviewer #3: Yes

3. Have the authors made all data underlying the findings in their manuscript fully available?

Reviewer #1: Yes

Reviewer #2: Yes

Reviewer #3: Yes

4. Is the manuscript presented in an intelligible fashion and written in standard English?

Reviewer #1: Yes

Reviewer #2: Yes

Reviewer #3: No

Reviewer #1: In the submitted manuscript entitled “Computational Identification of Epifriedelanol and Derived Analogs from Mikania cordata as Potential HMG-CoA Reductase Inhibitors”, the authors virtually screened a total of 92 phytocompounds of this medicinal species targeting the HMG-CoA reductase protein. As well, the authors performed molecular docking computations and molecular dynamics simulations over 500 ns. The manuscript is not well-presented, and the results are not well-discussed. Further technical details regarding the molecular docking calculations must be presented. The performance of the employed docking technique must be evaluated before any calculations based on experimental data. Full technical details regarding molecular docking computations and molecular dynamics simulations should be provided in the revised manuscript. All figures are poorly presented. Consequently, the manuscript in its current form is not suitable for publication in PLOS ONE.

Comments:

#########

Comment #0: The manuscript must be revised, where some typos and grammatical errors were observed.

Comment #1: The introduction provides a general context for the study. However, it does not explain the premises of the work and, more specifically, the rationale of the study. As well, the results should not be positioned in the introduction section.

Comment #2: Could the authors clarify the rationale for selecting this older PDB entry, despite the availability of more recent structures with improved resolution?

Comment #3: Why did the authors utilize CASTp for identifying the binding site, although the binding site of this protein is already known?

Comment #4: What is the exhaustiveness number utilized for performing molecular docking computations?

Comment #5: The authors claimed that “the ADMET evaluation was conducted using SwissADME to assess the drug-likeness properties of the selected compounds”. However, there is a big difference between ADMET and drug likeness properties. Please revise and correct.

Comment #6: Full technical details regarding molecular dynamics simulations should be provided in the revised manuscript.

Comment #7: How did the authors compute the atomic charges of the investigated compounds prior to molecular dynamics simulations?

Comment #8: Table 1 is very long. It should be summarized, and other data transported into supporting information.

Comment #9: Molecular interactions of the predicted docking poses of the identified analogs should be detailed in the revised manuscript.

Comment #10: What is the benefit of computing binding energy after docking calculations?

Comment #11: The number of tables is very huge. The non-informative tables should be transported into the supporting information.

Comment #12: Post-MD calculations should include binding energy per frame, center-of-mass, and hydrogen bond analyses..., etc.

Comment #13: The binding energy calculations over MD simulations should be computed using the MM-GBSA or MM-PBSA approach in the revised manuscript.

Comment #14: The performance of the employed docking technique must be evaluated before any calculations based on experimental data.

Comment #15: The number of figures is very large in the current manuscript; non-informative figures should be transported into the supporting information.

Comment #16: RMSF estimates the apo-receptor and ligand-soaked complex variance throughout the simulation time. However, the data are presented in Figure 13.

Comment #17: According to the post-MD analyses, the investigated compounds are not stable in the binding site of the investigated protein.

Comment #18: The quality of the presented figures must be improved.

Comment #19: Negative sign must be replaced by a minus sign in all manuscripts.

Comment #20: Reference style must be revised. There are versatile omitted details.

Comment #21: The conclusion section must be rewritten to be more informative and unveil the most beneficial outcomes.

Comment #22: All docking scores and binding energies should be in one decimal unit.

Comment #23: The number of keywords should be decreased to at most five.

Comment #24: Did the authors investigate the protonation state of the titrable amino acids before docking?

Comment #25: Did the authors investigate the protonation state of the ligands before docking computations?

Comment #26: Full technique details regarding molecular docking computations should be considered in the revised manuscript, such as the grid spacing value.

Comment #27: The reference of the utilized pdb code should be cited in the revised manuscript.

Comment #28: Did the authors perform equilibration for the investigated complexes prior to executing molecular dynamics simulations?

Reviewer #2: Reviewer Comments

The manuscript titled “Computational Identification of Epifriedelanol and Derived Analogs from Mikania cordata as Potential HMG-CoA Reductase Inhibitors”

In general, this is an interesting research topic approach, and a useful dataset has been provided. However, there are many important major points that must be addressed before considered for acceptance.

1. In Abstract, abbreviations that are used for the first time should be written in full name. (For Ex. HMG-CoA, ADMET, MM/GBSA….) Rewrite the Abstract.

2. From which part of the Mikania cordata plant were the 92 phytocompounds extracted?

3. In Introduction part a short explanation on the traditional applications of Mikania cordata in cardiovascular / metabolic disorders would help bridge ethnomedicine with modern pharmacology.

4. Cite recent references (Example: 10.1371/journal.pone.0325613, DOI:10.1371/journal.pone.0323702)

5. A statement of the novelty of the manuscript has to be added at the end of the introduction.

6. Provide in separate Supplementary Table the structures of these 92 selected compounds,

7. In Table 1, kindly mention the phytochemical names (From IMPHY014842 to IMPHY011562)

8. Check the abbreviations throughout the manuscript (Ex. Line 221 The ADME, Table 3 ADME, Make it ADMET, ….. check carefully)

9. Docking/simulation software versions, parameters should be cited properly

10. Clear and consistent figure is essential for readability and scientific accuracy.

11. Please provide good resolution figure with complete labelling. (For Ex. in Fig 9A label on X axis Time (nsec) font size is not same as Fig 9B). Please check for all figures X / Y-axis and maintain uniformity for all figures.

12. Figures are many. Some figures could be moved to supplementary file.

13. There are some typing mistakes as well, and authors are advised to carefully proof-read the text.

14. The computational results are promising, the manuscript would benefit from a clearer outline of future experimental plans. Suggestions for in vitro assays and in vivo models would enhance translational relevance.

15. Modify conclusion part. A clearer link between results and practical implications would strengthen the conclusion.

16. Few references are not as per Journal format. The authors have to check the consistency of the reference format in the text as per the journal format.

Reviewer #3: The manuscript titled “Computational Identification of Epifriedelanol and Derived Analogs from Mikania cordata as Potential HMG-CoA Reductase Inhibitors” presents an extensive in silico screening and molecular dynamics-based approach to identify plant-derived inhibitors of HMG-CoA reductase. While the study is relevant and methodologically ambitious, the current version of the manuscript is not suitable for publication due to incomplete methodological details, weak presentation of results, and several technical and formatting inconsistencies. The overall computational workflow is acceptable, but the lack of methodological validation, figure quality, and precise scientific justification limits the study’s reproducibility and impact. The overall computational workflow is acceptable, but the lack of methodological validation, figure quality, and precise scientific justification limits the study’s reproducibility and impact.

Major Issues

• The molecular docking methodology lacks essential parameters such as grid box dimensions, exhaustiveness level, and scoring function.

• The molecular dynamics (MD) section omits force field justification, charge assignment methods, and equilibration steps.

• The authors should clearly state how atomic charges were computed for ligands and whether the system was equilibrated prior to production runs.

• The rationale for selecting the specific PDB entry (1HWK) must be clarified—newer, higher-resolution structures exist.

• Before docking experimental data, the performance of the docking setup should be evaluated (e.g., cross-docking or known inhibitor validation).

• The post-MD analysis is incomplete. Include MM-GBSA/MM-PBSA binding free energy calculations over simulation trajectories, center-of-mass analysis, hydrogen bond persistence, and stability metrics.

• Figures are of poor quality and lack clarity; non-informative or redundant visuals should be moved to the Supporting Information.

• Several tables are excessively long (e.g., Table 1) and should be summarized.

• Numerous grammatical and typographical errors are present.

• Replace all hyphens (“-”) with proper minus signs (–).

• References are incomplete and inconsistent with PLOS ONE style.

• Limit keywords to a maximum of five.

• The RMSF and RMSD analyses suggest some complexes are not stable. These results should be discussed critically instead of claiming high stability.

• The benefits and limitations of computing binding energy post-docking must be explained.

• Clarify the difference between ADMET and “drug-likeness” — these are not synonymous.

• Rewrite the conclusion to focus on the most significant results and implications for future in vitro validation. Avoid repetition of results.

• Cite the reference for the PDB structure used.

• Include details of ligand and receptor protonation states prior to docking.

• Ensure consistency of docking and energy units to one decimal place.

• Improve figure legends for clarity and readability.

**Do you want your identity to be public for this peer review?** For information about this choice, including consent withdrawal, please see our Privacy Policy

Reviewer #1: No

Reviewer #2: No

Reviewer #3: No

---

## [Author Response · Author response to Decision Letter 1]

14 Nov 2025

Editorial Comments

Comment 1. Please ensure that your manuscript meets PLOS ONE's style requirements, including those for file naming. The PLOS ONE style templates can be found at

Response 1. We have thoroughly revised the manuscript and all associated files to ensure full compliance with PLOS ONE’s formatting and style requirements, including file naming conventions, manuscript structure, and consistent referencing throughout the text.

Comment 2. Please note that PLOS One has specific guidelines on code sharing for submissions in which author-generated code underpins the findings in the manuscript. In these cases, we expect all author-generated code to be made available without restrictions upon publication of the work. Please review our guidelines at https://journals.plos.org/plosone/s/materials-and-software-sharing#loc-sharing-code and ensure that your code is shared in a way that follows best practice and facilitates reproducibility and reuse.

Response 2. In accordance with PLOS ONE’s guidelines for reproducibility and open science, all author-generated code underlying the findings of this study has been made publicly available in the Zenodo repository and can be accessed via the following DOI: https://doi.org/10.5281/zenodo.17553513

Comment 3. We note that your Data Availability Statement is currently as follows: “All relevant data are available within the manuscript”

Response 3. All raw data necessary to replicate the results of this study have been made publicly available. These data are provided both as Supporting Information accompanying the revised manuscript and in a stable, open-access repository (Zenodo) at https://doi.org/10.5281/zenodo.17553513

Comment 4. If the reviewer comments include a recommendation to cite specific previously published works, please review and evaluate these publications to determine whether they are relevant and should be cited. There is no requirement to cite these works unless the editor has indicated otherwise.

Response 4. We thank the reviewer for the suggested citations. After careful evaluation, we found the recommended works relevant to our study and have incorporated them appropriately in the revised manuscript.

Additional Editor Comments

If reviewers recommend citations that are not relevant to the manuscript, authors are encouraged to decline them. The journal assures that choosing not to include such irrelevant citations will not affect the editorial decision on the manuscript.

Response. Reviewer 2 has recommended two papers for citations. We have found them relevant and cited in the revised manuscript. Thank you.

Reviewer 1

Comment 0: The manuscript must be revised, where some typos and grammatical errors were observed.

Response 0. Thank you for your suggestion. The manuscript has been thoroughly revised, and all typographical and grammatical errors have been corrected.

Comment 1: The introduction provides a general context for the study. However, it does not explain the premises of the work and, more specifically, the rationale of the study. As well, the results should not be positioned in the introduction section.

Response 1. We sincerely thank the reviewer for the insightful suggestion. In the revised manuscript, we have clarified the premises and rationale of the study by explicitly explaining the existing knowledge gap concerning the absence of systematic investigations on M. cordata phytoconstituents as HMG-CoA reductase inhibitors. The Introduction now emphasizes the scientific reasoning for selecting M. cordata, highlighting its well-documented traditional medicinal uses, broad-spectrum bioactivities, and pharmacological relevance to cardiovascular health.

Additionally, we have carefully rephrased all statements in the Introduction to focus on study objectives rather than outcomes, thereby avoiding any implication of results being reported in this section. These revisions ensure that the Introduction now clearly presents the rationale, objective, and novelty of the study in alignment with the journal’s guidelines.

Comment 2: Could the authors clarify the rationale for selecting this older PDB entry, despite the availability of more recent structures with improved resolution?

Response 2. We thank the reviewer for this insightful comment. The PDB structure 1HWK (Istvan and Deisenhofer, 2001) was selected because it represents the first and most extensively validated X-ray crystal structure of human HMG-CoA reductase complexed with a statin inhibitor. With a resolution of 2.22 Å, 1HWK has served as a benchmark model in numerous molecular docking, inhibitor design, and molecular dynamics studies over the past two decades (Ali et al., 2025; Bano et al., 2025; Suhail et al., 2025). Its longstanding and consistent use in literature supports its structural reliability and reproducibility for computational analyses.

We are aware that more recent structures, such as 8PKN (released January 2025; Cryo-EM, 2.26 Å) and 8S6B (released February 2025; Cryo-EM, 2.06 Å) (Karuppasamy and van Rooyen, 2025). However, both models were determined by single-particle Cryo-EM, exhibit shorter sequence coverage (423 amino acids), and represent different aggregation states (particle-based reconstructions). In contrast, 1HWK provides a complete 467-residue catalytic domain obtained via high-quality X-ray diffraction, with well-defined refinement statistics (R-Value Work = 0.212, R-Value Free = 0.235).

Therefore, despite the recent availability of Cryo-EM structures with comparable or slightly improved nominal resolution, 1HWK remains the most comprehensive, experimentally validated, and widely cited crystal structure of human HMG-CoA reductase-statin complex. This makes it the most appropriate and consistent choice for structure-based molecular docking and inhibitor interaction analyses in the present study. The rationale for this selection has been discussed in the Discussion section and is highlighted in yellow in the revised manuscript.

Ali, S., Ali, U., & Jan, M. I. (2025). Hypolipidemic and HMG-CoA reductase inhibitory effects of 15-oxoursolic acid from Rhododendron arboreum: An in-vivo and in-silico study. Drug Development and Industrial Pharmacy, 1-12.

Bano, S., Ansari, J. A., Ahsan, F., & Khan, A. R. (2025). Molecular docking, phytochemical screening and evaluation of Balliospermum montanum roots for its antihyperlipidemic potential. Cell Biochemistry and Biophysics, 1-18.

Istvan, E. S., & Deisenhofer, J. (2001). Structural mechanism for statin inhibition of HMG-CoA reductase. Science, 292(5519), 1160-1164.

Karuppasamy, M., & van Rooyen, J. (2025). Cryo-EM structures of apo and atorvastatin-bound human 3-hydroxy-3-methylglutaryl-coenzyme A reductase. Structural Biology and Crystallization Communications, 81(3), 118-122.

Suhail, P., Joy, J., Musliyarakath, N., Padmaraj, C. P., Theruvath, A., & Varghese, N. (2025). Bioactive compounds from Carica papaya (Papaya) fruit peel target key enzymes in diabetes and dyslipidemia: A molecular insight. Biomedical and Biotechnology Research Journal, 9(3), 246-254.

Comment 3: Why did the authors utilize CASTp for identifying the binding site, although the binding site of this protein is already known?

Response 3. We sincerely thank the reviewer for this valuable comment. Although the catalytic site of HMG-CoA reductase complexed with statins (e.g., Atorvastatin) is well-established, several recent studies have identified additional druggable pockets and regulatory interfaces within this enzyme (PDB ID 1HWK). Therefore, in the present study, we employed the CASTp server to identify all possible surface-accessible cavities and to evaluate the likelihood of alternative or allosteric binding regions for the selected natural compounds.

Our CASTp-based prediction revealed a prominent cavity encompassing residues such as Tyr517, Ile531, Met534, and Pro813, which partially overlaps with residues reported as critical for dimer formation and stability by Gesto et al. (2014), including Ile531 and Met534. Furthermore, Kanu et al. (2024) demonstrated that natural green tea polyphenols (EGCG, ECG, EC) bind to Glu528, Met534, Ile536, and Lys735, distinct from the canonical Atorvastatin site, yet exhibit strong inhibitory potential against HMG-CoA reductase.

In light of these findings, the CASTp-derived pocket was considered a biologically meaningful and experimentally supported region for ligand docking, enabling the exploration of potential alternative inhibition mechanisms of the enzyme beyond the classical catalytic site. Notably, our results showed that both the control (Atorvastatin) and the phytochemicals bound within the same CASTp-predicted cavity, reinforcing the reliability of our docking setup. We have discussed the reason of using CASTp for identifying the binding site, highlighted in yellow, in the Discussion section.

Gesto, D. S., Cerqueira, N. M. F. S. A., Ramos, M. J., & Fernandes, P. A. (2014). Discovery of new druggable sites in the anti-cholesterol target HMG-CoA reductase by computational alanine scanning mutagenesis. Journal of Molecular Modeling, 20(4), 2178.

Kanu, V. R., Pulakuntla, S., Kuruvalli, G., Aramgam, S. L., Marthadu, S. B., Pannuru, P., ... & Vaddi, D. R. (2024). Anti-atherogenic role of green tea (Camellia sinensis) in South Indian smokers. Journal of Ethnopharmacology, 332, 118298.

Comment 4: What is the exhaustiveness number utilized for performing molecular docking computations?

Response 4. We thank the reviewer for this query. The exhaustiveness parameter in EasyDock Vina v.2.237 was set to 9, which is the fixed default value in the software. This value ensures sufficient sampling of the ligand conformational space during docking. We have explicitly added this detail explicitly in the subsection “2.4. Molecular docking” of the revised manuscript.

Comment 5: The authors claimed that “the ADMET evaluation was conducted using SwissADME to assess the drug-likeness properties of the selected compounds”. However, there is a big difference between ADMET and drug likeness properties. Please revise and correct.

Response 5. Thank you very much for your critical observation. We fully agree that ADMET and drug-likeness represent distinct aspects of compound evaluation. Accordingly, we have revised the Methodology section to clarify this distinction. The updated text now reads:

“The pharmacokinetic and drug-likeness properties of the selected compounds were evaluated using the SwissADME server [45]. Subsequently, toxicity parameters were predicted separately using the STopTox server to assess potential adverse effects, completing the comprehensive ADMET evaluation [46].”

This revision ensures that the methodological description accurately reflects the separate evaluation of pharmacokinetic, drug-likeness, and toxicity properties.

Comment 6: Full technical details regarding molecular dynamics simulations should be provided in the revised manuscript.

Response 6. Thank you for the suggestion. In the revised manuscript, the MD simulation methodology [subsection 2.8 Molecular Dynamics (MD) Simulation] has been substantially expanded to include complete technical details, such as ensemble conditions, thermostat and barostat controls, long-range electrostatics treatment, sampling frequency, and post-MD analytical procedures. The revised methodology part now comprehensively describes the simulation setup, ensuring transparency and reproducibility.

Comment 7: How did the authors compute the atomic charges of the investigated compounds prior to molecular dynamics simulations?

Response 7. We thank the reviewer for raising this important point. As described in the subsection 2.8, partial atomic charges for all ligands were automatically assigned during system preparation in Maestro, ensuring full compatibility with the OPLS4 force-field parameters used in Desmond. This procedure ensures physically consistent charge representation throughout the simulation workflow.

Comment 8: Table 1 is very long. It should be summarized, and other data transported into supporting information.

Response 8. Thank you for your valuable suggestion. We have shortened Table 1 by retaining only the top 24 phytocompounds considering the binding affinely of over -6.0 kcal/mol, while the complete dataset comprising all 91 phytocompounds along with the control drug Atorvastatin has been moved to the supplementary section (Table S1). Additionally, following the recommendation of Reviewer 2, two new columns (Compound name and Plant part used) have been incorporated into both Table 1 and Table S1.

Comment 9: Molecular interactions of the predicted docking poses of the identified analogs should be detailed in the revised manuscript.

Response 9. Thank you for your constructive suggestion. In response, the molecular interaction profiles of the identified analogs have been elaborated in detail in the revised manuscript (Section 3.5.2. Molecular interaction analysis of structural analogs of Epifriedelanol). The updated section now provides a comprehensive description of the docking poses of EA2 and EA3, including the specific binding site residues, interaction types (pi–alkyl and carbon–hydrogen interactions), and corresponding bond distances.

Comment 10: What is the benefit of computing binding energy after docking calculations?

Response 10. We thank the reviewer for the comment regarding the benefit of computing binding energy after docking. Post-docking MM/GBSA calculations provide a quantitative estimate of ligand-protein binding affinity, enabling efficient prioritization of potential leads. In our study, this approach allowed us to identify Epifriedelanol as the most promising phytocompound (having the best free binding energy) from Mikania cordata (ΔG_bind = −52.3 kcal/mol), while other candidates such as Taraxasterol (having inferior free binding energy than Epifriedelanol) were eliminated despite favorable ADMET profiles (Table S2). Structural analogs EA2 and EA3 were then shortlisted based on their superior binding energies (−59.8 and −59.7 kcal/mol) relative to the control drug Atorvastatin (−44.7 kcal/mol). This workflow is consistent with previous studies; for example, Shimu et al. (2022) employed MM/GBSA immediately after docking to select promising phytochemical inhibitors of Dengue NS2B/NS3 protein, and Ahmed et al. (2022) used a similar approach to prior

---

## [Decision Letter · Decision Letter 1]

25 Nov 2025

Dear Dr. Rahman,

Thank you for submitting your manuscript to PLOS ONE. After careful consideration, we feel that it has merit but does not fully meet PLOS ONE’s publication criteria as it currently stands. Therefore, we invite you to submit a revised version of the manuscript that addresses the points raised during the review process.

We look forward to receiving your revised manuscript.

Kind regards,

Chandrabose Selvaraj, Ph.D.

Academic Editor

PLOS ONE

Journal Requirements:

Reviewers' comments:

Reviewer's Responses to Questions

**Comments to the Author**

Reviewer #1: (No Response)

Reviewer #2: All comments have been addressed

Reviewer #3: All comments have been addressed

2. Is the manuscript technically sound, and do the data support the conclusions?

Reviewer #1: (No Response)

Reviewer #2: Yes

Reviewer #3: Yes

3. Has the statistical analysis been performed appropriately and rigorously?

Reviewer #1: (No Response)

Reviewer #2: Yes

Reviewer #3: Yes

4. Have the authors made all data underlying the findings in their manuscript fully available?

Reviewer #1: (No Response)

Reviewer #2: Yes

Reviewer #3: Yes

5. Is the manuscript presented in an intelligible fashion and written in standard English?

Reviewer #1: (No Response)

Reviewer #2: Yes

Reviewer #3: Yes

Reviewer #1: In the revised version of the submitted manuscript entitled “Computational Identification of Epifriedelanol and Derived Analogs from Mikania cordata as Potential HMG-CoA Reductase Inhibitors”, most of our comments were taken into account. Nevertheless, the comments listed below are still ambigious and must be solved. Accordingly, a minor revision is a must before approving the acceptance of the manuscript in the “PLOS ONE” journal.

Comments:

#########

Comment #1: As shown in Figure 1, the authors report an RMSD of 2.163 Å between the predicted docking pose and the experimental pose. However, it is well-established in the scientific literature that an RMSD value below 2.0 Å is generally required to validate a docking protocol as accurate and reliable. Therefore, the reported RMSD exceeds the commonly accepted threshold, raising concerns about the robustness of the docking validation presented in the manuscript.

Comment #2: All abbreviations utilized in the abstract should be defined.

Comment #3: Full equations utilized for computing binding energy using the MM-GBSA approach over MD simulations should be mentioned in the revised manuscript.

Comment #4: 100 ps is not sufficient for the equilibration stage of the investigated complexes nowadays. It should be at least 10 ns.

Comment #5: What is the justification for a temperature of 300 K?

Comment #6: Again, RMSF estimates the apo-receptor and ligand-soaked complex variance throughout the simulation time. However, the data are presented in Figure 6 that did not show the data of the apo-structure.

Comment #7: Why did the authors perform the equilibration stage in the NVT conditions?

Comment #8: As illustrated in Figure 9, Atorvastatin shows positive binding energy values beyond 300 ns of MD simulation. These positive values imply a loss of binding affinity and indicate that Atorvastatin may no longer interact with the protein between 300 and 500 ns. Further explanation is required.

Reviewer #2: The authors have responded well to the comments. The manuscript has improved in both scientific clarity and presentation. To further strengthen the manuscript, I suggest the following minor key points to be addressed to meet the journal’s formatting and quality expectations.

1. Please ensure consistency in figure labeling throughout the manuscript. For example, at line 523 it is written as 'Figs S5 and S6,' at line 564 as 'Fig 8A–D,' and elsewhere as 'Fig.8E.' Kindly maintain a uniform style for figure references across the text."

2. In results section line number 582, in the sentence 'epifriedelanol, EA2, EA3, and atorvastatin demonstrated mean free binding energies of −39.5, 583 −31.3, −43.7, and −21.4 kcal/mol, respectively,' Should be corrected as 'Epifriedelanol, EA2, EA3, and atorvastatin demonstrated mean free binding energies of −39.5, 583 −31.3, −43.7, and −21.4 kcal/mol, respectively. Please carefully check the entire MS for similar typographical or formatting errors to ensure consistency and clarity."

3. Please avoid repeating the full species name 'Mikania cordata' throughout the MS. After the first mention, you may abbreviate it as M. cordata. (Ex. line no. 101, 288,340, etc)

Reviewer #3: In the revised manuscript entitled " Computational Identification of Epifriedelanol and Derived Analogs from Mikania cordata as Potential HMG-CoA Reductase Inhibitors", the authors considered all our comments. Consequently, the manuscript may be published in the Journal of PLOS ONE.

**Do you want your identity to be public for this peer review?** For information about this choice, including consent withdrawal, please see our Privacy Policy

Reviewer #1: No

Reviewer #2: No

Reviewer #3: No

---

## [Author Response · Author response to Decision Letter 2]

22 Dec 2025

EDITORIAL COMMENTS

Comment 1: If the reviewer comments include a recommendation to cite specific previously published works, please review and evaluate these publications to determine whether they are relevant and should be cited. There is no requirement to cite these works unless the editor has indicated otherwise.

Response 1: We thank the editor for this note. In the present review, the reviewer did not recommend citation of any specific previously published works. Therefore, no additional evaluation or citation was required.

Comment 2: Please review your reference list to ensure that it is complete and correct. If you have cited papers that have been retracted, please include the rationale for doing so in the manuscript text, or remove these references and replace them with relevant current references. Any changes to the reference list should be mentioned in the rebuttal letter that accompanies your revised manuscript. If you need to cite a retracted article, indicate the article’s retracted status in the References list and also include a citation and full reference for the retraction notice.

Response 2: We have thoroughly reviewed the entire reference list to ensure its accuracy, completeness, and compliance with PLOS ONE referencing guidelines. We confirm that no retracted articles were cited in the manuscript. During this revision, two additional references were added to support our response to Comment 1 of Reviewer 1, and these references have been formatted following PLOS ONE guidelines. The reference list and in-text citations were updated accordingly to ensure consistency and to avoid any anomalies. All changes to the references have been reflected in the revised manuscript.

REVIEWER 1

In the revised version of the submitted manuscript entitled “Computational Identification of Epifriedelanol and Derived Analogs from Mikania cordata as Potential HMG-CoA Reductase Inhibitors”, most of our comments were taken into account. Nevertheless, the comments listed below are still ambigious and must be solved. Accordingly, a minor revision is a must before approving the acceptance of the manuscript in the “PLOS ONE” journal.

Comment 1: As shown in Figure 1, the authors report an RMSD of 2.163 Å between the predicted docking pose and the experimental pose. However, it is well-established in the scientific literature that an RMSD value below 2.0 Å is generally required to validate a docking protocol as accurate and reliable. Therefore, the reported RMSD exceeds the commonly accepted threshold, raising concerns about the robustness of the docking validation presented in the manuscript.

Response 1: We thank the reviewer for raising the important point regarding the RMSD threshold used for docking validation. While an RMSD value of ≤ 2.0 Å is often cited as a benchmark for good pose reproduction, it is not a strict or universally enforced cutoff. Rather, RMSD values should be interpreted in the context of ligand flexibility, receptor rigidity, and conservation of key binding interactions.

Several well-cited studies have demonstrated that RMSD values slightly above 2.0 Å can still represent valid and reliable docking solutions. For example, Ramírez and Caballero (2018) categorized docking poses with RMSD values between 2.0 and 3.0 Å as acceptable solutions that preserve the correct ligand orientation and key binding features. Similarly, Wu et al. (2020) reported a redocking RMSD of 2.1129 Å for the co-crystal ligand of human maltase-glucoamylase (PDB ID: 3TOP) and concluded that the docking strategy was valid, emphasizing that despite RMSD > 2.0 Å, the docked and crystallographic poses were well superimposed at hydrogen-bonding sites. In our study, the obtained RMSD of 2.163 Å falls within this accepted range and is comparable to values reported in the above-mentioned validated docking studies. We have clarified this point in the revised Discussion section and added the relevant references (highlighted yellow) to improve transparency and address the reviewer’s concern.

Ramírez, D., & Caballero, J. (2018). Is it reliable to take the molecular docking top scoring position as the best solution without considering available structural data?. Molecules, 23(5), 1038.

Wu, J., Hu, B., Sun, X., Wang, H., Huang, Y., Zhang, Y., ... & Yu, Z. (2020). In silico study reveals existing drugs as α-glucosidase inhibitors: Structure-based virtual screening validated by experimental investigation. Journal of Molecular Structure, 1218, 128532.

Comment 2: All abbreviations utilized in the abstract should be defined.

Response 2: Thank you. We have ensured that all abbreviations are clearly defined upon their first occurrence.

Comment 3: Full equations utilized for computing binding energy using the MM-GBSA approach over MD simulations should be mentioned in the revised manuscript.

Response 3: Thank you for your valuable suggestion. In response, we have now explicitly included the full equation for computing binding energy using the MM-GBSA approach over MD simulations at section “2.9. Post-MD simulation MM/GBSA analysis” in the revised manuscript. These additions are clearly indicated and highlighted in yellow for ease of identification.

Comment 4: 100 ps is not sufficient for the equilibration stage of the investigated complexes nowadays. It should be at least 10 ns.

Response 4: We thank the reviewer for this important comment regarding the equilibration duration. We would like to clarify that the 100 ps simulation mentioned in the manuscript refers specifically to the initial Brownian dynamics stage performed under NVT conditions at 10 K with restrained solute heavy atoms, which represents only the first step of a multi-stage equilibration protocol, rather than the entirety of the the equilibration process. Subsequent stages included short NVT and NPT simulations with restraints, followed by unrestrained NPT equilibration, ensuring gradual relaxation of solvent and protein-ligand interactions prior to the production phase.

Similar multi-step equilibration strategies with short initial simulations are widely used in contemporary molecular dynamics studies and are considered standard practice in the literature, including publications in PLOS ONE. For example, Mohamed et al. (2024, PLOS ONE) employed a 100 ps Brownian dynamics stage as the first step in a seven-stage equilibration protocol prior to a 200 ns production run. Similarly, Naidoo et al. (2025) and Zhao et al. (2025) applied initial 100 ps Brownian dynamics steps as part of their multi-stage equilibration protocols. These studies demonstrate that short initial equilibration steps, when combined with staged equilibration and followed by sufficiently long production simulations, are adequate to achieve thermodynamic stability.

Mohamed, G. A., Abdallah, H. M., Sindi, I. A., Ibrahim, S. R., & Alzain, A. A. (2024). Unveiling the potential of phytochemicals to inhibit nuclear receptor binding SET domain protein 2 for cancer: Pharmacophore screening, molecular docking, ADME properties, and molecular dynamics simulation investigations. PLoS ONE, 19(8), e0308913.

Naidoo, V., Achilonu, I., Mirza, S., Hull, R., Kandhavelu, J., Soobben, M., & Penny, C. (2025). Computational Modelling of Tunicamycin C Interaction with Potential Protein Targets: Perspectives from Inverse Docking with Molecular Dynamic Simulation. Current Issues in Molecular Biology, 47(5), 339.

Zhao, R., Hou, L., Tesfagaber, W., Song, L., Zhang, Z., Li, F., ... & Zhao, D. (2025). Virtual Screening and Molecular Dynamics Simulation Targeting the ATP Domain of African Swine Fever Virus Type II DNA Topoisomerase. Viruses, 17(5), 681.

Comment 5: What is the justification for a temperature of 300 K?

Response 5: Thank you for this pertinent question. The simulation temperature of 300 K was chosen to approximate physiological conditions for human HMG-CoA reductase while maintaining computational stability. This temperature is widely adopted in the literature for MD simulations of HMG-CoA reductase-ligand complexes to mimic a biologically relevant environment and to ensure realistic atomic motions. For example, Mazumder et al. (2024, Aspects of Molecular Medicine), Khori et al. (2023, Chemical Physics Impact), and Son et al. (2013, PLOS ONE) all conducted MD simulations of HMG-CoA reductase complexes at 300 K, maintaining stable trajectories and reliable conformational sampling. These studies demonstrate that 300 K provides a physiologically relevant temperature suitable for capturing dynamic behavior, protein-ligand interactions, and thermodynamic stability over the course of long simulations.

Khori, V., Zahedi, M., Mirzaei, H., Jabbari, A., & Hosseini, S. G. (2023). Repurposing of monacolin K decorated BN nanoparticle on inhibition of HMG-CoA reductase: In silico approach. Chemical Physics Impact, 7, 100384.

Mazumder, R., Choudhury, D., Sarkar, A., Ghosh, A., Debnath, S., Debnath, B., & Ghosh, R. (2024). In silico approach for identification of potential tetracyclic triterpenoids from mushroom as HMG-CoA reductase inhibitor. Aspects of Molecular Medicine, 4, 100053.

Son, M., Baek, A., Sakkiah, S., Park, C., John, S., & Lee, K. W. (2013). Exploration of virtual candidates for human HMG-CoA reductase inhibitors using pharmacophore modeling and molecular dynamics simulations. PLoS ONE, 8(12), e83496.

Comment 6: Again, RMSF estimates the apo-receptor and ligand-soaked complex variance throughout the simulation time. However, the data are presented in Figure 6 that did not show the data of the apo-structure.

Response 6: Thank you for this valuable suggestion. In response, we have revised Figure 6 to include both RMSD and RMSF profiles of the apoprotein over the 500 ns MD simulation alongside the ligand-bound complexes. Specifically, the RMSD and RMSF profiles of the apoprotein are now shown in the revised Figs 6E and 6F, respectively. The Results section has been updated accordingly to reflect these additions.

Comment 7: Why did the authors perform the equilibration stage in the NVT conditions?

Response 7: We thank the reviewer for this insightful question. The initial NVT equilibration was employed to stabilize the system’s temperature while restraining the protein and ligand heavy atoms. During this stage, the number of particles (N), system volume (V), and temperature (T) are kept constant, allowing the solvent molecules to relax around the solute without large volume fluctuations that could distort the protein-ligand complex.

After the temperature was stabilized under NVT conditions, we switched to NPT equilibration, allowing the system to adjust its pressure and density under a constant number of particles, pressure, and temperature. This stepwise NVT → NPT equilibration is a widely accepted standard practice in molecular dynamics simulations (e.g., Mohamed et al., 2024, PLOS ONE; Zhao et al., 2025, Viruses), ensuring gradual relaxation of both solvent and solute and preventing structural artifacts before the production run.

Mohamed, G. A., Abdallah, H. M., Sindi, I. A., Ibrahim, S. R., & Alzain, A. A. (2024). Unveiling the potential of phytochemicals to inhibit nuclear receptor binding SET domain protein 2 for cancer: Pharmacophore screening, molecular docking, ADME properties, and molecular dynamics simulation investigations. PLoS ONE, 19(8), e0308913.

Zhao, R., Hou, L., Tesfagaber, W., Song, L., Zhang, Z., Li, F., ... & Zhao, D. (2025). Virtual Screening and Molecular Dynamics Simulation Targeting the ATP Domain of African Swine Fever Virus Type II DNA Topoisomerase. Viruses, 17(5), 681.

Comment 8: As illustrated in Figure 9, Atorvastatin shows positive binding energy values beyond 300 ns of MD simulation. These positive values imply a loss of binding affinity and indicate that Atorvastatin may no longer interact with the protein between 300 and 500 ns. Further explanation is required.

Response 8: We thank the reviewer for this insightful observation regarding the MM/GBSA binding free energy profile of atorvastatin. The transient appearance of positive ΔG_bind values for atorvastatin beyond ~300 ns does not necessarily indicate complete ligand dissociation. Instead, these values reflect increased conformational flexibility and a relative weakening of interaction strength compared to the lead compounds during the later stages of the simulation.

MM/GBSA binding free energy calculations are highly sensitive to instantaneous conformational fluctuations, solvent exposure, and changes in protein-ligand interaction geometry, particularly for flexible ligands such as atorvastatin. Consequently, short-lived positive ΔG_bind values may arise even when the ligand remains within the binding pocket, especially during periods of partial disengagement or increased mobility rather than full unbinding.

This interpretation is supported by ligand-centric stability analyses indicating that atorvastatin remained associated with the active site throughout the simulation. The protein-ligand center-of-mass (PL-COM) distance did not show any abrupt or sustained increase indicative of complete ligand dissociation, confirming continued spatial proximity to the catalytic pocket (Fig 8E). Additionally, hydrogen-bond analysis revealed that although the number of hydrogen bonds decreased during the later stages of the simulation, intermittent interactions persisted until the end of the simulation trajectory (Fig 8D). Together, these observations indicate that the positive MM/GBSA values observed after 300 ns reflect a weakening of binding interactions of atorvastatin rather than a complete loss of protein-ligand association. This clarification has been added to the Discussion section of the revised manuscript (highlighted in yellow).

REVIEWER 2

The authors have responded well to the comments. The manuscript has improved in both scientific clarity and presentation. To further strengthen the manuscript, I suggest the following minor key points to be addressed to meet the journal’s formatting and quality expectations.

Comment 1: Please ensure consistency in figure labeling throughout the manuscript. For example, at line 523 it is written as 'Figs S5 and S6,' at line 564 as 'Fig 8A–D,' and elsewhere as 'Fig.8E.' Kindly maintain a uniform style for figure references across the text."

Response 1: Thank you for pointing out the inconsistency in figure labeling. We have carefully revised the manuscript to ensure a uniform style throughout. Specifically, the dot after “Fig.” has been removed, “Fig 8A–D” has been standardized to “Figs 8A–D,” and “Fig. 8E” has been corrected to “Fig 8E.” All figure references now follow a consistent formatting style across the entire manuscript.

Comment 2: In results section line number 582, in the sentence 'epifriedelanol, EA2, EA3, and atorvastatin demonstrated mean free binding energies of −39.5, 583 −31.3, −43.7, and −21.4 kcal/mol, respectively,' Should be corrected as 'Epifriedelanol, EA2, EA3, and atorvastatin demonstrated mean free binding energies of −39.5, 583 −31.3, −43.7, and −21.4 kcal/mol, respectively. Please carefully check the entire MS for similar typographical or formatting errors to ensure consistency and clarity."

Response 2: Thank you very much for your critical observation. We have amended the anomaly and thoroughly revised the manuscript. All similar inconsistencies have been checked and corrected throughout the manuscript, highlighted in yellow.

Comment 3: Please avoid repeating the full species name 'Mikania cordata' throughout the MS. After the first mention, you may abbreviate it as M. cordata. (Ex. line no. 101, 288,340, etc).

Response 3: Thank you for the helpful suggestion. We have revised the manuscript to use the abbreviated form M. cordata after its first full mention. All subsequent occurrences throughout the text (including those at lines 101, 288, 340, and elsewhere) have been updated accordingly to maintain consistency and avoid repetition. All the changes have been highlighted in yellow.

REVIEWER 3

Comment: In the revised manuscript entitled "Computational Identification of Epifriedelanol and Derived Analogs from Mikania cordata as Potential HMG-CoA Reductase Inhibitors", the authors considered all our comments. Consequently, the manuscript may be published in the Journa

---

## [Editor Report · Decision Letter 2]

23 Dec 2025

Computational Identification of Epifriedelanol and Derived Analogs from Mikania cordata as Potential HMG-CoA Reductase Inhibitors

PONE-D-25-51503R2

Dear Dr. Rahman,

We’re pleased to inform you that your manuscript has been judged scientifically suitable for publication and will be formally accepted for publication once it meets all outstanding technical requirements.

Kind regards,

Chandrabose Selvaraj, Ph.D.

Academic Editor

PLOS One
---

## [Editor Report · Acceptance letter]

PONE-D-25-51503R2

PLOS One

Dear Dr. Rahman,

I'm pleased to inform you that your manuscript has been deemed suitable for publication in PLOS One. Congratulations! Your manuscript is now being handed over to our production team.

Kind regards,

on behalf of

Dr. Chandrabose Selvaraj

Academic Editor

PLOS One